# NEURAL CDEs FOR LONG TIME SERIES VIA THE LOG-ODE METHOD

## ABSTRACT

Neural Controlled Differential Equations (Neural CDEs) are the continuous-time analogue of an RNN, just as Neural ODEs are analogous to ResNets. However just like RNNs, training Neural CDEs can be difficult for long time series. Here, we propose to apply a technique drawn from stochastic analysis, namely the log-ODE method. Instead of using the original input sequence, our procedure summarises the information over local time intervals via the log-signature map, and uses the resulting shorter stream of log-signatures as the new input. This represents a length/channel trade-off. In doing so we demonstrate efficacy on problems of length up to 17k observations and observe significant training speed-ups, improvements in model performance, and reduced memory requirements compared to the existing algorithm.

## 1 INTRODUCTION

Neural controlled differential equations (Neural CDEs) (Kidger et al., 2020) are the continuous-time analogue to a recurrent neural network (RNN), and provide a natural method for modelling temporal dynamics with neural networks.

Neural CDEs are similar to neural ordinary differential equations (Neural ODEs), as popularised by Chen et al. (2018). A Neural ODE is determined by its initial condition, without a direct way to modify the trajectory given subsequent observations. In contrast the vector field of a Neural CDE depends upon the time-varying data, so that the trajectory of the system is driven by a sequence of observations.

### 1.1 CONTROLLED DIFFERENTIAL EQUATIONS

We begin by stating the definition of a CDE.

Let $a, b \in \mathbb{R}$ with $a < b$, and let $v, w \in \mathbb{N}$. Let $\xi \in \mathbb{R}^w$. Let $X : [a, b] \to \mathbb{R}^v$ be a continuous function of bounded variation (which is for example implied by it being Lipschitz), and let $f : \mathbb{R}^w \to \mathbb{R}^{w \times v}$ be continuous.

Then we may define $Z : [a, b] \to \mathbb{R}^w$ as the unique solution of the *controlled differential equation*

$$Z_a = \xi, \qquad Z_t = Z_a + \int_a^t f(Z_s)\mathrm{d}X_s \quad \text{for } t \in (a, b], \tag{1}$$

The notation "$f(Z_s)\mathrm{d}X_s$" denotes a matrix-vector product, and if $X$ is differentiable then $\int_a^t f(Z_s)\mathrm{d}X_s = \int_a^t f(Z_s)\frac{\mathrm{d}X}{\mathrm{d}s}(s)\mathrm{d}s$.

If in equation (1), $\mathrm{d}X_s$ was replaced with $\mathrm{d}s$, then the equation would just be an ODE. Using $\mathrm{d}X_s$ causes the solution to depend continuously on the evolution of $X$. We say that the solution is "driven by the control, $X$".

### 1.2 NEURAL CONTROLLED DIFFERENTIAL EQUATIONS

We recall the definition of a Neural CDE as introduced in Kidger et al. (2020).

Consider a time series $\mathbf{x}$ as a collection of points $x_i \in \mathbb{R}^{v-1}$ with corresponding time-stamps $t_i \in \mathbb{R}$ such that $\mathbf{x} = ((t_0, x_0), (t_1, x_1), ..., (t_n, x_n))$, and $t_0 < ... < t_n$.

Let $X : [t_0, t_n] \to \mathbb{R}^v$ be some interpolation of the data such that $X_{t_i} = (t_i, x_i)$. Kidger et al. (2020) use natural cubic splines. Here we will actually end up finding piecewise linear interpolation to be a more convenient choice. (We avoid issues with adaptive solvers as discussed in Kidger et al. (2020, Appendix A) simply by using fixed solvers.)

Let $\xi_\theta : \mathbb{R}^v \to \mathbb{R}^w$ and $f_\theta : \mathbb{R}^w \to \mathbb{R}^{w \times v}$ be neural networks. Let $\ell_\theta : \mathbb{R}^w \to \mathbb{R}^q$ be linear, for some output dimension $q \in \mathbb{N}$. Here $\theta$ is used to denote dependence on learnable parameters.

We define $Z$ as the hidden state and $Y$ as the output of a *neural controlled differential equation* driven by $X$ if

$$Z_{t_0} = \xi_\theta(t_0, x_0), \quad \text{with} \quad Z_t = Z_{t_0} + \int_{t_0}^t f_\theta(Z_s) \mathrm{d}X_s \quad \text{and} \quad Y_t = \ell_\theta(Z_t) \quad \text{for } t \in (t_0, t_n]. \quad (2)$$

That is – just like an RNN – we have evolving hidden state $Z$, which we take a linear map from to produce an output. This formulation is a universal approximator (Kidger et al., 2020, Appendix B). The output may be either the time-evolving $Y_t$ or just the final $Y_{t_n}$. This is then fed into a loss function ($L^2$, cross entropy, ...) and trained via stochastic gradient descent in the usual way.

The question remains how to compute the integral of equation (2). Kidger et al. (2020) let

$$g_{\theta,X}(Z, s) = f_\theta(Z) \frac{\mathrm{d}X}{\mathrm{d}s}(s), \quad (3)$$

where the right hand side denotes a matrix multiplication, and then note that the integral can be written as

$$Z_t = Z_{t_0} + \int_{t_0}^t g_{\theta,X}(Z_s, s) \mathrm{d}s. \quad (4)$$

This reduces the CDE to an ODE, so that existing tools for Neural ODEs may be used to evaluate this, and to backpropagate.

By moving from the discrete-time formulation of an RNN to the continuous-time formulation of a Neural CDE, then every kind of time series data is put on the same footing, whether it is regularly or irregularly sampled, whether or not it has missing values, and whether or not the input sequences are of consistent length.

Besides this, the continuous-time or differential equation formulation may be useful in applications where such models are explicitly desired, as when modelling physics.

### 1.3 CONTRIBUTIONS

Neural CDEs, as with RNNs, begin to break down for long time series. Training loss/accuracy worsens, and training time becomes prohibitive due to the sheer number of forward operations within each training epoch.

Here, we apply the *log-ODE method*, which is a numerical method from stochastic analysis and rough path theory. It is a method for converting a CDE to an ODE, which may in turn be solved via standard ODE solvers. Thus this acts as a drop-in replacement for the original procedure that uses the derivative of the control path.

In particular, we find that this method is particularly beneficial for long time series (and incidentally does not require differentiability of the control path). With this method both training time and model performance of Neural CDEs are improved, and memory requirements are reduced.

The resulting scheme has two very neat interpretations. In terms of numerical differential equation solvers, this corresponds to taking integration steps larger than the discretisation of the data, whilst incorporating substep information through additional terms[1]. In terms of machine learning, this corresponds to binning the data prior to running a Neural CDE, with bin statistics carefully chosen to extract precisely the information most relevant to solving a CDE.

---

[1]For the reader familiar with numerical methods for SDEs, this is akin to the additional correction term in Milstein's method as compared to Euler-Maruyama.

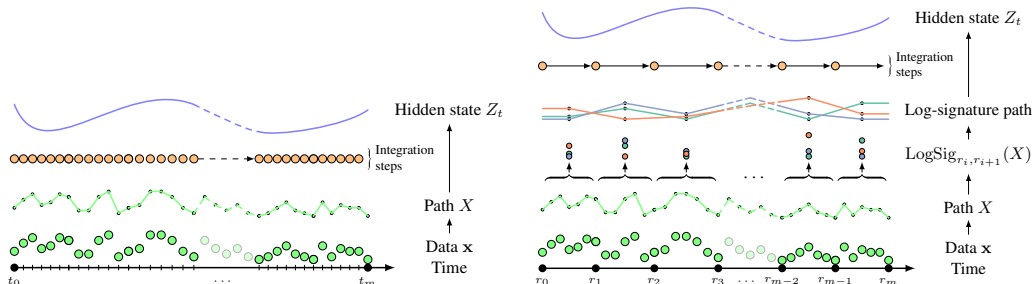

Figure 1: **Left:** The original Neural CDE formulation. The path $X$ is quickly varying, meaning a lot of integration steps are needed to resolve it. **Right:** The log-ODE method. The log-signature path is more slowly varying (in a higher dimensional space), and needs fewer integration steps to resolve.

## 2 THEORY

We begin with motivating theory, though we note that this section is not essential for using the method. Readers more interested in practical applications should feel free to skip to section 3.

### 2.1 SIGNATURES AND LOG-SIGNATURES

The signature transform is a map from paths to a vector of real values, specifying a collection of statistics about the path. It is a central component of the theory of controlled differential equations, since these statistics describe how the data interacts with dynamical systems. The log-signature is then formed by representing the same information in a compressed format.

We begin by providing a formal definition of the signature, and a description of the log-signature. We will then give some intuition, first into the geometry of the first few terms of the (log-)signature, and then by providing a short example of how these terms appear when solving CDEs.

**Signature transform** Let $\mathbf{x} = (x_1, ..., x_n)$, where $x_i \in \mathbb{R}^v$. Let $T > 0$ and $0 = t_1 < t_2 < ... < t_{n-1} < t_n = T$ be arbitrary. Let $X = (X^1, ..., X^d) : [0, T] \to \mathbb{R}^d$ be the unique continuous function such that $X(t_i) = x_i$ and is affine on the intervals between (essentially just a linear interpolation of the data). Letting[2]

$$S_{a,b}^{i_1, \cdots i_k}(X) = \int \cdots \int_{0 < t_1 < \ldots < t_k < T} \prod_{j=1}^{k} \frac{\mathrm{d}X^{i_j}}{\mathrm{d}t}(t_j) \mathrm{d}t_j, \tag{5}$$

then the depth-N signature transform of $X$ is given by

$$\mathrm{Sig}_{a,b}^N(X) = \left( \left\{ S(X)^{(i)} \right\}_{i=1}^d, \left\{ S(\mathbf{x})^{(i,j)} \right\}_{i,j=1}^d, \ldots, \left\{ S(\mathbf{x})^{(i_1, ..., i_N)} \right\}_{i_1, ..., i_N=1}^d \right). \tag{6}$$

This definition is independent of the choice of $T$ and $t_i$ (Bonnier et al., 2019, Proposition A.7).

We see that the signature is a collection of integrals, with each integral defining a real value. It is a graded sequence of statistics that characterise the input time series. In particular, (Hambly & Lyons, 2010) show that under mild conditions, $\mathrm{Sig}^\infty(X)$ completely determines $X$ up to translation (provided time is included in a channel in $X$).

**Log-signature transform** However, the signature transform has some redundancy: a little algebra shows that for example $S_{a,b}^{1,2}(X) + S_{a,b}^{2,1}(X) = S_{a,b}^1(X)S_{a,b}^2(X)$, so that for instance we already know $S_{a,b}^{2,1}(X)$ provided we know the other three quantities.

---

[2]This is a slightly simplified definition, and the signature is often instead defined using the notation of stochastic calculus; see Definition A.2.

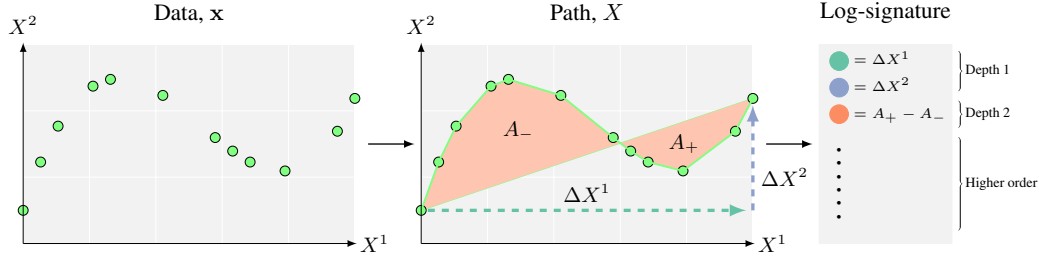

Figure 2: Geometric intuition for the first two levels of the log-signature for a 2-dimensional path. The depth 1 terms correspond to the change in each of the coordinates over the interval. The depth 2 term corresponds to the *Lévy area* of the path, this being the signed area between the curve and the chord joining its start and endpoints.

The *log-signature transform* is then essentially obtained by computing the signature transform, and throwing out redundant terms, to obtain some (nonunique) minimal collection.

Starting from the depth-N signature transform and removing some fixed set of redundancies produces the *depth-N log-signature transform*.[3] We denote this $\mathrm{LogSig}_{a,b}^N$, which is a map from Lipschitz continuous paths $[a, b] \to \mathbb{R}^v$ into $\mathbb{R}^{\beta(v,N)}$, where $\beta(v, N)$ denotes the dimension of the log-signature. The precise procedure is a little involved; both this and a formula for $\beta(v, N)$ can be found in Appendix A.

**Geometric intuition** In figure 2 we provide a geometric intuition for the first two levels of the log-signature (which have particularly natural interpretations).

**(Log-)Signatures and CDEs** (Log-)signatures are intrinsically linked to solutions of CDEs. Let $D_f$ denote the Jacobian of a function $f$. Now expand equation (1) by linearising the vector field $f$ and neglecting higher order terms:

$$
\begin{aligned}
Z_t &\approx Z_a + \int_a^t \Big( f(Z_a) + D_f(Z_a)(Z_s - Z_a) \Big) \frac{\mathrm{d}X}{\mathrm{d}t}(s) \mathrm{d}s \\
&= Z_a + \int_a^t \left( f(Z_a) + D_f(Z_a) \int_a^s f(Z_u) \frac{\mathrm{d}X}{\mathrm{d}t}(u)\, \mathrm{d}u \right) \frac{\mathrm{d}X}{\mathrm{d}t}(s)\, \mathrm{d}s \\
&\approx Z_a + f(Z_a) \int_a^t \frac{\mathrm{d}X}{\mathrm{d}t}(s)\, \mathrm{d}s + D_f(Z_a)f(Z_a) \int_a^t \int_a^s \frac{\mathrm{d}X}{\mathrm{d}t}(u)\, \mathrm{d}u \frac{\mathrm{d}X}{\mathrm{d}t}(s) \mathrm{d}s \\
&= Z_a + f(Z_a)\big\{ S(X)^{(i)} \big\}_{i=1}^d + D_f(Z_a)f(Z_a)\big\{ S(X)^{(i,j)} \big\}_{i,j=1}^d.
\end{aligned}
\tag{7}
$$

This gives a Taylor expansion of the solution, and moreover the coefficients involve the terms in the signature. Higher order Taylor expansions results in corrections using higher order signature terms. We refer the reader to section 7.1 of Friz & Victoir (2010) for further details.

## 2.2 THE LOG-ODE METHOD

Recall for $X : [a, b] \to \mathbb{R}^v$ that $\mathrm{LogSig}_{a,b}^N(X) \in \mathbb{R}^{\beta(v,N)}$. The log-ODE method states

$$
Z_b \approx \widehat{Z}_b \quad \text{where} \quad \widehat{Z}_u = \widehat{Z}_a + \int_a^u \widehat{f}(\widehat{Z}_s) \frac{\mathrm{LogSig}_{a,b}^N(X)}{b - a} \mathrm{d}s, \quad \text{and} \quad \widehat{Z}_a = Z_a.
\tag{8}
$$

where $Z$ is as defined in equation (2), and the relationship between $\widehat{f}$ to $f$ is given in Appendix A.

That is, the solution of the CDE may be approximated by the solution to an ODE. In practice, we go further and pick some points $r_i$ such that $a = r_0 < r_1 < \cdots < r_m = b$. We split up the CDE of

---

[3]Similar terminology such as "step-N log-signature" is also used in the literature.

equation (1) into an integral over $[r_0, r_1]$, an integral over $[r_1, r_2]$, and so on, and apply the log-ODE method to each interval separately.

See Appendix A for more details and Appendix B for a proof of convergence.

Also see Janssen (2011); Lyons (2014); Boutaib et al. (2014) for other discussions of the log-ODE method. See Gaines & Lyons (1997); Gyurkó & Lyons (2008); Flint & Lyons (2015); Foster et al. (2020) for applications of the log-ODE method to stochastic differential equations (SDEs).

## 3 METHOD

We move on to discussing the application of the log-ODE method to Neural CDEs.

Recall that we observe some time series $\mathbf{x} = ((t_0, x_0), (t_1, x_1), ..., (t_n, x_n))$, and have constructed a piecewise linear interpolation $X \colon [t_0, t_n] \to \mathbb{R}^v$ such that $X_{t_i} = (t_i, x_i)$.

We now pick points $r_i$ such that $t_0 = r_0 < r_1 < \cdots < r_m = t_n$. In principle these can be variably spaced but in practice we will typically space them equally far apart. The total number of points $m$ should be much smaller than $n$.

In section 2 the log-signature transform was introduced. To recap, for $X \colon [t_0, t_n] \to \mathbb{R}^v$ and $t_0 \leq r_i < r_{i+1} \leq t_n$ the depth-$N$ log-signature of $X$ over the interval $[r_i, r_{i+1}]$ is some collection of statistics $\mathrm{LogSig}_{r_i, r_{i+1}}^N(X) \in \mathbb{R}^{\beta(v, N)}$.

In particular, these statistics are precisely those most relevant for solving the CDE equation (1).

### 3.1 UPDATING THE NEURAL CDE HIDDEN STATE EQUATION VIA THE LOG-ODE METHOD

Recall how the Neural CDE formulation of equation (2) was solved via equations (3), (4). For the log-ODE approach we replace (3) with the piecewise

$$\widehat{g}_{\theta, X}(Z, s) = \widehat{f}_\theta(Z) \frac{\mathrm{LogSig}_{r_i, r_{i+1}}^N(X)}{r_{i+1} - r_i} \quad \text{for } s \in [r_i, r_{i+1}), \tag{9}$$

where $\widehat{f}_\theta \colon \mathbb{R}^w \to \mathbb{R}^{w \times \beta(v, N)}$ is an arbitrary neural network, and the right hand side denotes a matrix-vector product between $\widehat{f}_\theta$ and the log-signature. Equation (4) then becomes

$$Z_t = Z_{t_0} + \int_{t_0}^t \widehat{g}_{\theta, X}(Z_s, s) \mathrm{d}s. \tag{10}$$

This may now be solved as a (neural) ODE using standard ODE solvers.

### 3.2 RELATIONSHIP TO THE ORIGINAL METHOD

Suppose we happened to choose $r_i = t_i$ and $r_{i+1} = t_{i+1}$. Then the log-signature term is

$$\frac{\mathrm{LogSig}_{t_i, t_{i+1}}^N(X)}{t_{i+1} - t_i} \tag{11}$$

The depth 1 the log-signature is just the increment of the path over the interval, and so this becomes

$$\frac{\Delta X_{[t_i, t_{i+1}]}}{t_{i+1} - t_i} = \frac{\mathrm{d}X^{linear}}{\mathrm{d}t}(t_i) \quad \text{for } s \in [t_i, t_{i+1}), \tag{12}$$

that is to say the same as obtained via the original method if using linear interpolation.

### 3.3 DISCUSSION

**Ease of Implementation** This method is straightforward to implement using pre-existing tools. There are standard libraries available for computing the log-signature transform; we use Signatory (Kidger & Lyons, 2020b). Then, as equation (10) is an ODE, it may be solved directly using tools such as `torchdiffeq` (Chen, 2018).

As an alternative, we note that the formulation in equation (11) can be written in precisely the same form as equation (3), with the driving path taken to be piecewise linear in log-signature space. Computation of the log-signatures can therefore be considered as a preprocessing step, producing a sequence of logsignatures. From this we may construct a path in log-signature space, and apply existing tools for neural CDEs. This idea is summarised in figure 1. We make this approach available in the [redacted] open source project.

**Structure of $\widehat{f}$**   The description here aligns with the log-ODE scheme described in equation (8). There is one discrepancy: we do not attempt to model the specific structure of $\widehat{f}$. This is in principle possible, but is computationally expensive. Instead, we model $\widehat{f}$ as a neural network directly. This need *not* necessarily exhibit the requisite structure, but as neural networks are universal approximators (Pinkus, 1999; Kidger & Lyons, 2020a) then this approach is at least as general from a modelling perspective.

**Lossy Representation**   The log-signature transform can be thought of as a lossy representation for time series. This is made rigorous in Diehl et al. (2020), where it is shown that the log-signature can be obtained by iterating an "area" operation between paths. For CDEs, these geometric features precisely encode the interaction between the data and the system.

**Length/Channel Trade-Off**   The sequence of log-signatures is now of length $m$, which was chosen to be much smaller than $n$. As such, it is much more slowly varying over the interval $[t_0, t_n]$ than the original data, which was of length $n$. The differential equation it drives is better behaved, and so larger integration steps may be used in the numerical solver. This is the source of the speed-ups of this method; we observe typical speed-ups by a factor of about 100.

Each element is a log-signature of size $\beta(v, N) \geq v$; the additional channels are higher-order corrections to compensate for the larger integration steps.

**Generality of the Log-ODE Method**   If depth $N = 1$ and steps $r_i = t_i$ are used, then the above formulation exactly reduces onto the original Neural CDE formulation using linear interpolation. Thus the log-ODE method in fact generalises the original approach.

**Applications**   In principle the log-ODE method may be applied to solve any Neural CDE. In practice, the reduction in length (from $n$ to $m$), coupled with the loss of information (from using the log-signature as a summary statistic) makes this particularly useful for long time series.

**Memory Efficiency**   Long sequences need large amounts of memory to perform backpropagation-through-time (BPTT). As with the original Neural CDEs, the log-ODE approach supports memory-efficient backpropagation via the adjoint equations, alleviating this issue. See Kidger et al. (2020).

**The Depth and Step Hyperparameters**   To solve a Neural CDE accurately via the log-ODE method, we should be prepared to take the depth $N$ suitably large, or the intervals $r_{i+1} - r_i$ suitably small. Accomplishing this would realistically require that they are taken very large or very small, respectively. Instead, we treat these as hyperparameters. This makes use of the log-ODE method a modelling choice rather than an implementation detail.

Increasing step size will lead to faster (but less informative) training by reducing the number of operations in the forward pass. Increasing depth will lead to slower (but more informative) training, as more information about each local interval is used in each update.

## 4   EXPERIMENTS

We investigate solving a Neural CDE with and without the log-ODE method on four real-world problems. Every problem was chosen for its long length. The lengths are in fact sufficiently long that adjoint-based backpropagation (Chen et al., 2018) was needed to avoid running out of memory at any reasonable batch size. Every problem is regularly sampled, so we take $t_i = i$.

| Model | Step | Test Accuracy (%) | Time (Hrs) | Memory (Mb) |
|---|---|---|---|---|
| NCDE₁ | 1 | $62.4 \pm 12.1$ | 22.0 | 176.5 |
| | 8 | $64.1 \pm 13.3$ | 3.1 | 24.3 |
| | 32 | $64.1 \pm 14.3$ | 0.5 | 8.0 |
| | 128 | $48.7 \pm 2.6$ | 0.1 | 3.9 |
| NCDE₂ | 8 | $\mathbf{77.8 \pm 5.9}$ | 2.1 | 94.2 |
| | 32 | $67.5 \pm 12.1$ | 0.7 | 28.1 |
| | 128 | $\mathbf{76.1 \pm 5.9}$ | 0.2 | 7.8 |
| NCDE₃ | 8 | $70.1 \pm 6.5$ | 1.3 | 460.7 |
| | 32 | $\mathbf{75.2 \pm 3.0}$ | 0.6 | 134.7 |
| | 128 | $68.4 \pm 8.2$ | 0.1 | 53.3 |

Table 1: Mean and standard deviation of test set accuracy (in %) over three repeats, as well as memory usage and training time, on the EigenWorms dataset for depths 1–3 and a small selection of step sizes. The bold values denote that the model was the top performer for that step size.

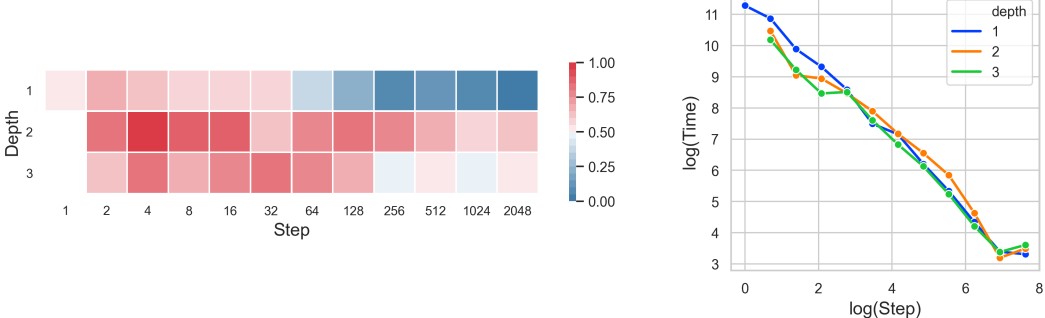

Figure 3: **Left:** Heatmap of normalised accuracies on the EigenWorms dataset for differing step sizes and depths. **Right:** Log-log plot of the elapsed time of the algorithm against the step size.

We will denote a Neural CDE model with log-ODE method, using depth $N$ and step $s$, as $\text{NCDE}_N^s$. Taking $N = 1$ (and any $s$) corresponds to not using the log-ODE method, with the data subsampled at rate $1/s$, as per section 3.3. Thus we use $\text{NCDE}_1^1$ as our benchmark: no subsampling, no log-ODE method.

In principle we could compare against RNN variants. This is for simple practical reasons: RNN-based models do not fit in the memory of the GPU resources we have available. This is one of the main advantages of using differential equation models in the first place, for which adjoint backpropagation is available. (As per the first paragraph of this section.)

Each model is run three times and we report the mean and standard deviation of the test metrics along with the mean training times and memory usages.

For each task, the hyperparameters were selected by performing a grid search on the $\text{NCDE}_1^s$ model, where $s$ was chosen so that the length of the sequence was 500 steps. This was found to create a reasonable balance between training time and sequence length. (Doing hyperoptimisation on the baseline $\text{NCDE}_1^1$ model would have been more difficult due to the larger training times.)

Precise details of the experiments can be found in Appendices C and D.

## 4.1 CLASSIFYING EIGENWORMS

Our first example uses the EigenWorms dataset from the UEA archive from Bagnall et al. (2017). This consists of time series of length 17 984 and 6 channels (including time), corresponding to the movement of a roundworm. The goal is to classify each worm as either wild-type or one of four mutant-type classes.

| Depth | Step | $L^2$ | | | Time (H) | | | Memory (Mb) |
|---|---|---|---|---|---|---|---|---|
| | | RR | HR | SpO$_2$ | RR | HR | SpO$_2$ | |
| NCDE$_1$ | 1 | $2.79 \pm 0.04$ | $9.82 \pm 0.34$ | $2.83 \pm 0.27$ | 23.8 | 22.1 | 28.1 | 56.5 |
| | 8 | $2.80 \pm 0.06$ | $10.72 \pm 0.24$ | $3.43 \pm 0.17$ | 3.0 | 2.6 | 4.8 | 14.3 |
| | 32 | $2.53 \pm 0.23$ | $12.23 \pm 0.43$ | $2.68 \pm 0.12$ | 1.9 | 0.9 | 2.2 | 9.8 |
| | 128 | $2.64 \pm 0.18$ | $11.98 \pm 0.37$ | $2.86 \pm 0.04$ | 0.2 | 0.2 | 0.3 | 8.7 |
| NCDE$_2$ | 8 | $2.63 \pm 0.12$ | $8.63 \pm 0.24$ | $2.88 \pm 0.15$ | 2.1 | 3.4 | 3.3 | 21.8 |
| | 32 | $1.90 \pm 0.02$ | $7.90 \pm 1.00$ | $1.69 \pm 0.20$ | 1.2 | 1.1 | 2.0 | 13.1 |
| | 128 | $1.86 \pm 0.03$ | $6.77 \pm 0.42$ | $1.95 \pm 0.18$ | 0.3 | 0.4 | 0.7 | 10.9 |
| NCDE$_3$ | 8 | $\mathbf{2.42 \pm 0.19}$ | $\mathbf{7.67 \pm 0.40}$ | $\mathbf{2.55 \pm 0.13}$ | 2.9 | 3.2 | 3.1 | 43.3 |
| | 32 | $\mathbf{1.67 \pm 0.01}$ | $\mathbf{4.50 \pm 0.70}$ | $\mathbf{1.61 \pm 0.05}$ | 1.3 | 1.8 | 7.3 | 20.5 |
| | 128 | $\mathbf{1.51 \pm 0.08}$ | $\mathbf{2.97 \pm 0.45}$ | $\mathbf{1.37 \pm 0.22}$ | 0.5 | 1.7 | 1.7 | 17.3 |

Table 2: Mean and standard deviation of the $L^2$ losses on the test set for each of the vitals signs prediction tasks (RR, HR, SpO$_2$) on the BIDMC dataset, across three repeats. Only mean times are shown for space. The memory usage is given as the mean over all three of the tasks as it was approximately the same for any task for a given depth and step. The bold values denote the algorithm with the lowest test set loss for a fixed step size for each task.

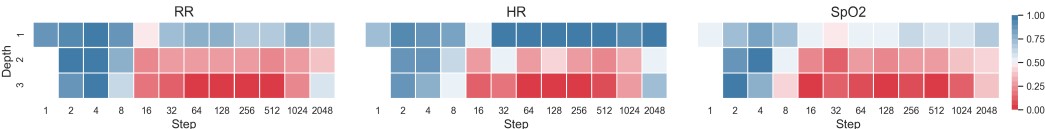

Figure 4: Heatmap of normalised losses on the three BIDMC datasets for differing step sizes and depths.

See Table 1. We see that the straightforward NCDE$_1^1$ model takes roughly a day to train. Using the log-ODE method (NCDE$_2$, NCDE$_3$) speeds this up to take roughly minutes. Doing so additionally improves model performance dramatically, and reduces memory usage. Naive subsampling approaches (NCDE$_1^8$, NCDE$_1^{32}$, NCDE$_1^{128}$) only achieve speed-ups without performance improvements, this can be seen in the NCDE$_1$ column which corresponds to naive subsampling for a step size greater than 1.

We notice that the NCDE$_3$ model has faster training times than the depth 2 model (and sometimes better then depth 1) over each step size. This is due to the fact we imposed a stopping criterion if the loss failed to decrease after 60 epochs, meaning that the NCDE$_3$ has converged with less epochs (time per epoch will still be larger though).

See also Figure 3, in which we summarise results for a larger range of step sizes.

## 4.2 ESTIMATING VITALS SIGNS FROM PPG AND ECG DATA

Next we consider the problem of estimating vital signs from PPG and ECG data. This comes from the TSR archive (Tan et al., 2020) using data from the Beth Israel Deaconess Medical Centre (BIDMC). We consider three separate tasks, in which we aim to predict a person's respiratory rate (RR), their heart rate (HR), and their oxygen saturation (SpO2). This data is sampled at 125Hz with each series having a length of 4 000. There are 7 949 training samples, and 3 channels (including time).

We train a model on each of the three vitals sign prediction tasks. The metric used to evaluate performance is the $L^2$ loss. The results over a range of step sizes are presented in table (2). We also provide heatmaps in Figure 4 for each dataset containing the loss values (normalised to $[0, 1]$) for each task. The full results over all step sizes may be found in Appendix D.

We find that the depth 3 model is the top performer for every task at any step size. What's more, it does so with a significantly reduced training time. We attribute the improved performance to

the log-ODE model being better able to learn long-term dependencies due to the reduced sequence length. Note that the performance of the $NCDE_2^s$, $NCDE_3^s$ models actually improves as the step size is increased. This is in contrast to $NCDE_1^s$, which sees a degradation in performance.

## 5 Limitations of the Log-ODE Method

**Number of hyperparameters**   Two new hyperparameters – truncation depth and step size – with substantial effects on training time and memory usage must now also be tuned.

**Number of input channels**   The log-ODE method is most feasible for low numbers of input channels, as the number of log-signature channels $\beta(v, N)$ grows exponentially in $v$.

## 6 Related Work

There has been some work on long time series for classic RNN (GRU/LSTM) models.

Wisdom et al. (2016); Jing et al. (2019) show that unitary or orthogonal RNNs can mitigate the vanishing/exploding gradients problem. However, they are expensive to train due to the need to compute a matrix inversion at each training step. Chang et al. (2017) introduce dilated RNNs with skip connections between RNN states, which help improve training speed and learning of long-term dependencies. Campos et al. (2017) introduce the 'Skip-RNN' model, which extend the RNN by adding an additional learnt component that skips state updates. Li et al. (2018) introduce the 'IndRNN' model, with particular structure tailored to learning long time series.

One important comparison is to hierarchical subsampling as in Graves (2012); De Mulder et al. (2015). There the data is split into windows, an RNN is run over each window, and then an additional RNN is run over the first RNN's outputs; we may describe this as an RNN/RNN pair. Liao et al. (2019) then perform the equivalent operation with a log-signature/RNN pair. In this context, our use of log-ODE method may then be described as a log-signature/NCDE pair.

In comparison to Liao et al. (2019), this means moving from an inspired choice of pre-processing to an actual implementation of the log-ODE method. In doing so the differential equation structure is preserved. Moreover this takes advantage of the synergy between log-signatures (which extract statistics on how data drives differential equations), and the controlled differential equation it then drives. Broadly speaking these connections are natural: at least within the signature/CDE/rough path community, it is a well-known but poorly-published fact that (log-)signatures, RNNs and (Neural) CDEs are all related; see for example Kidger et al. (2020) for a little exposition on this.

CNNs and Transformers have been shown to offer improvements over RNNs for modelling long-term dependencies (Bai et al., 2018; Li et al., 2019). However, both can be expensive in their own right; Transformers are famously $\mathcal{O}(L^2)$ in the length of the time series $L$. Whilst several approaches have been introduced to reduce this, for example Li et al. (2019) reduce this to $\mathcal{O}(L(\log L)^2)$, this can still be difficult with long series. Extensions specifically to long sequences do exist (Sourkov, 2018), but these typically focus on language modelling rather than multivariate time series data.

De Brouwer et al. (2019); Lechner & Hasani (2020) amongst others consider continuous time analogues of GRUs and LSTMs, going some way to improving the learning of long-term dependencies. Voelker et al. (2019); Gu et al. (2020) consider links with ODEs and approximation theory, to improve the long-term memory capacity of RNNs.

## 7 Conclusion

We demonstrate how to effectively apply Neural CDEs to long (17k) time series, via the log-ODE method. The model may still be solved via ODE methods and thus retains adjoint backpropagation and continuous dynamics. In doing so we see significant training speed-ups, improvements in model performance, and reduced memory requirements.

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

# Supplementary material

In sections A and B, we give a more thorough introduction to solving CDEs via the log-ODE method.

In section C we discuss the experimental details such as the choice of network structure, computing infrastructure and hyperparameter selection approach.

In section D we give a full breakdown of every experimental result.

## A  AN INTRODUCTION TO THE LOG-ODE METHOD FOR CONTROLLED DIFFERENTIAL EQUATIONS

The log-ODE method is an effective method for approximating the controlled differential equation:

$$\mathrm{d}Y_t = f(Y_t)\,\mathrm{d}X_t, \tag{13}$$
$$Y_0 = \xi,$$

where $X : [0, T] \to \mathbb{R}^d$ has finite length, $\xi \in \mathbb{R}^n$ and $f : \mathbb{R}^n \to L(\mathbb{R}^d, \mathbb{R}^n)$ is a function with certain smoothness assumptions so that the CDE (13) is well posed. Throughout these appendices, $L(U, V)$ denotes the space of linear maps between the vector spaces $U$ and $V$. In rough path theory, the function $f$ is referred to as the "vector field" of (13) and usually assumed to have $\mathrm{Lip}(\gamma)$ regularity (see definition 10.2 in Friz & Victoir (2010)). In this section, we assume one of the below conditions on the vector field:

1. $f$ is bounded and has $N$ bounded derivatives.
2. $f$ is linear.

In order to define the log-ODE method, we will first consider the tensor algebra and path signature.

**Definition A.1** *We say that* $T(\mathbb{R}^d) := \mathbb{R} \oplus \mathbb{R}^d \oplus (\mathbb{R}^d)^{\otimes 2} \oplus \cdots$ *is the* tensor algebra *of* $\mathbb{R}^d$ *and* $T((\mathbb{R}^d)) := \left\{ \boldsymbol{a} = (a_0, a_1, \cdots) : a_k \in (\mathbb{R}^d)^{\otimes k} \ \forall k \geq 0 \right\}$ *is the set of formal series of tensors of* $\mathbb{R}^d$. *Moreover,* $T(\mathbb{R}^d)$ *and* $T((\mathbb{R}^d))$ *can be endowed with the operations of addition and multiplication. Given* $\boldsymbol{a} = (a_0, a_1, \cdots)$ *and* $\boldsymbol{b} = (b_0, b_1, \cdots)$, *we have*

$$\boldsymbol{a} + \boldsymbol{b} = (a_0 + b_0, a_1 + b_1, \cdots), \tag{14}$$
$$\boldsymbol{a} \otimes \boldsymbol{b} = (c_0, c_1, c_2, \cdots), \tag{15}$$

*where for* $n \geq 0$, *the* $n$-*th term* $c_n \in (\mathbb{R}^d)^{\otimes n}$ *can be written using the usual tensor product as*

$$c_n := \sum_{k=0}^{n} a_k \otimes b_{n-k}.$$

*The operation* $\otimes$ *given by (15) is often referred to as the "tensor product".*

**Definition A.2** *The* signature *of a finite length path* $X : [0, T] \to \mathbb{R}^d$ *over the interval* $[s, t]$ *is defined as the following collection of iterated (Riemann-Stieltjes) integrals:*

$$S_{s,t}(X) := \left( 1, X_{s,t}^{(1)}, X_{s,t}^{(2)}, X_{s,t}^{(3)}, \cdots \right) \in T((\mathbb{R}^d)), \tag{16}$$

*where for* $n \geq 1$,

$$X_{s,t}^{(n)} := \int \cdots \int_{s < u_1 < \cdots < u_n < t} \mathrm{d}X_{u_1} \otimes \cdots \otimes \mathrm{d}X_{u_n} \in (\mathbb{R}^d)^{\otimes n}.$$

*Similarly, we can define the depth-N (or truncated) signature of the path* $X$ *on* $[s, t]$ *as*

$$S_{s,t}^N(X) := \left( 1, \int_{s < u_1 < t} \mathrm{d}X_u, \cdots, \int \cdots \int_{s < u_1 < \cdots < u_N < t} \mathrm{d}X_{u_1} \otimes \cdots \otimes \mathrm{d}X_{u_N} \right) \in T^N(\mathbb{R}^d), \tag{17}$$

*where* $T^N(\mathbb{R}^d) := \mathbb{R} \oplus \mathbb{R}^d \oplus (\mathbb{R}^d)^{\otimes 2} \oplus \cdots \oplus (\mathbb{R}^d)^{\otimes N}$ *denotes the truncated tensor algebra.*

The (truncated) signature provides a natural feature set that describes the effects a path $X$ has on systems that can be modelled by (13). That said, defining the log-ODE method actually requires the so-called "log-signature" which efficiently encodes the same integral information as the signature. The log-signature is obtained from the path's signature by removing certain algebraic redundancies, such as

$$\int_0^t \int_0^s \mathrm{d}X_u^i \mathrm{d}X_s^j + \int_0^t \int_0^s \mathrm{d}X_u^j \mathrm{d}X_s^i = X_t^i X_t^j,$$

for $i, j \in \{1, \cdots, d\}$, which follows by the integration-by-parts formula. To this end, we will define the logarithm map on the depth-$N$ truncated tensor algebra $T^N(\mathbb{R}^d) := \mathbb{R} \oplus \mathbb{R}^d \oplus \cdots \oplus (\mathbb{R}^d)^{\otimes N}$.

**Definition A.3 (The logarithm of a formal series)** *For $\boldsymbol{a} = (a_0, a_1, \cdots) \in T((\mathbb{R}^d))$ with $a_0 > 0$, define $\log(\boldsymbol{a})$ to be the element of $T((\mathbb{R}^d))$ given by the following series:*

$$\log(\boldsymbol{a}) := \log(a_0) + \sum_{n=1}^{\infty} \frac{(-1)^n}{n} \left(\boldsymbol{1} - \frac{\boldsymbol{a}}{a_0}\right)^{\otimes n}, \tag{18}$$

*where $\boldsymbol{1} = (1, 0, \cdots)$ is the unit element of $T((\mathbb{R}^d))$ and $\log(a_0)$ is viewed as $\log(a_0)\boldsymbol{1}$.*

**Definition A.4 (The logarithm of a truncated series)** *For $\boldsymbol{a} = (a_0, a_1, \cdots, a_N) \in T((\mathbb{R}^d))$ with $a_0 > 0$, define $\log^N(\boldsymbol{a})$ to be the element of $T^N(\mathbb{R}^d)$ defined from the logarithm map (18) as*

$$\log^N(\boldsymbol{a}) := P_N(\log(\widetilde{\boldsymbol{a}})), \tag{19}$$

*where $\widetilde{\boldsymbol{a}} := (a_0, a_1, \cdots, a_N, 0, \cdots) \in T((\mathbb{R}^d))$ and $P_N$ denotes the standard projection map from $T((\mathbb{R}^d))$ onto $T^N(\mathbb{R}^d)$.*

**Definition A.5** *The* log-signature *of a finite length path $X : [0, T] \to \mathbb{R}^d$ over the interval $[s, t]$ is defined as $\mathrm{LogSig}_{s,t}(X) := \log(S_{s,t}(X))$, where $S_{s,t}(X)$ denotes the path signature of $X$ given by Definition A.2. Likewise, the depth-N (or truncated) log-signature of $X$ is defined for each $N \geq 1$ as $\mathrm{LogSig}_{s,t}^N(X) := \log^N(S_{s,t}^N(X))$.*

The log-signature is a map from $X \colon [0, T] \to \mathbb{R}^d \to \mathbb{R}^{\beta(d,N)}$. The exact form of $\beta(d, N)$ is given by

$$\beta(d, N) = \sum_{k=1}^N \frac{1}{k} \sum_{i|k} \mu\left(\frac{k}{i}\right) d^i$$

with $\mu$ the Möbius function. We note that the order of this remains an open question.

The final ingredient we use to define the log-ODE method are the derivatives of the vector field $f$. It is worth noting that these derivatives also naturally appear in the Taylor expansion of (13).

**Definition A.6 (Vector field derivatives)** *We define $f^{\circ k} : \mathbb{R}^n \to L((\mathbb{R}^d)^{\otimes k}, \mathbb{R}^n)$ recursively by*

$$f^{\circ(0)}(y) := y,$$
$$f^{\circ(1)}(y) := f(y),$$
$$f^{\circ(k+1)}(y) := D(f^{\circ k})(y)f(y),$$

*for $y \in \mathbb{R}^n$, where $D(f^{\circ k})$ denotes the Fréchet derivative of $f^{\circ k}$.*

Using these definitions, we can describe two closely related numerical methods for the CDE (13).

**Definition A.7 (The Taylor method)** *Given the CDE (13), we can use the path signature of $X$ to approximate the solution $Y$ on an interval $[s, t]$ via its truncated Taylor expansion. That is, we use*

$$\mathrm{Taylor}(Y_s, f, S_{s,t}^N(X)) := \sum_{k=0}^N f^{\circ k}(Y_s)\pi_k(S_{s,t}^N(X)), \tag{20}$$

*as an approximation for $Y_t$ where each $\pi_k : T^N(\mathbb{R}^d) \to (\mathbb{R}^d)^{\otimes k}$ is the projection map onto $(\mathbb{R}^d)^{\otimes k}$.*

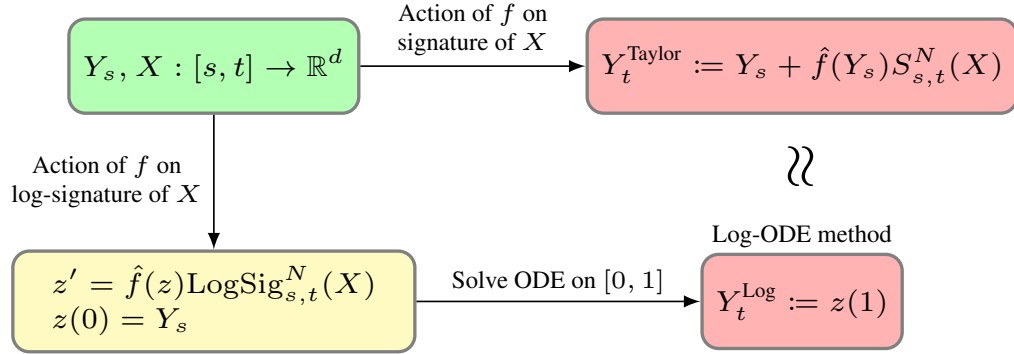

Figure 5: Illustration of the log-ODE and Taylor methods for controlled differential equations.

**Definition A.8 (The Log-ODE method)** *Using the Taylor method (20), we can define the function $\widehat{f} : \mathbb{R}^n \to L(T^N(\mathbb{R}^d), \mathbb{R}^n)$ by $\widehat{f}(z) := \mathrm{Taylor}(z, f, \cdot)$. By applying $\widehat{f}$ to the truncated log-signature of the path $X$ over an interval $[s, t]$, we can define the following ODE on $[0, 1]$*

$$\frac{\mathrm{d}z}{\mathrm{d}u} = \widehat{f}(z)\mathrm{LogSig}_{s,t}^N(X), \tag{21}$$

$$z(0) = Y_s.$$

*Then the log-ODE approximation of $Y_t$ (given $Y_s$ and $\mathrm{LogSig}_{s,t}^N(X)$) is defined as*

$$\mathrm{LogODE}(Y_s, f, \mathrm{LogSig}_{s,t}^N(X)) := z(1). \tag{22}$$

**Remark A.9** *Our assumptions of $f$ ensure that $z \mapsto \widehat{f}(z)\mathrm{LogSig}_{s,t}^N(X)$ is either globally bounded and Lipschitz continuous or linear. Hence both the Taylor and log-ODE methods are well defined.*

**Remark A.10** *It is well known that the log-signature of a path $X$ lies in a certain free Lie algebra (this is detailed in section 2.2.4 of Lyons et al. (2007)). Furthermore, it is also a theorem that the Lie bracket of two vector fields is itself a vector field which doesn't depend on choices of basis. By expressing $\mathrm{LogSig}_{s,t}^N(X)$ using a basis of the free Lie algebra, it can be shown that only the vector field $f$ and its (iterated) Lie brackets are required to construct the log-ODE vector field $\widehat{f}(z)\mathrm{LogSig}_{s,t}^N(X)$. In particular, this leads to our construction of the log-ODE (8) using the Lyndon basis of the free Lie algebra (see Reizenstein (2017) for a precise description of the Lyndon basis). We direct the reader to Lyons (2014) and Boutaib et al. (2014) for further details on this Lie theory.*

To illustrate the log-ODE method, we give two examples:

**Example A.11 (The "increment-only" log-ODE method)** *When $N = 1$, the ODE (21) becomes*

$$\frac{\mathrm{d}z}{\mathrm{d}u} = f(z)X_{s,t},$$

$$z(0) = Y_s.$$

*Therefore we see that this "increment-only" log-ODE method is equivalent to driving the original CDE (13) by a piecewise linear approximation of the control path $X$. This is a classical approach for stochastic differential equations (i.e. when $X_t = (t, W_t)$ with $W$ denoting a Brownian motion) and is an example of a Wong-Zakai approximation (see Wong & Zakai (1965) for further details).*

**Example A.12 (An application for SDE simulation)** *Consider the following affine SDE,*

$$\mathrm{d}Y_t = a(b - y_t)\,\mathrm{d}t + \sigma y_t \circ \mathrm{d}W_t, \tag{23}$$

$$y(0) = y_0 \in \mathbb{R}_{\geq 0},$$

where $a, b \geq 0$ are the mean reversion parameters, $\sigma \geq 0$ is the volatility and $W$ denotes a standard real-valued Brownian motion. The $\circ$ means that this SDE is understood in the Stratonovich sense. The SDE (23) is known in the literature as Inhomogeneous Geometric Brownian Motion (or IGBM). Using the control path $X = \{(t, W_t)\}_{t \geq 0}$ and setting $N = 3$, the log-ODE (21) becomes

$$\frac{\mathrm{d}z}{\mathrm{d}u} = a(b - z_u)h + \sigma\, z_u W_{s,t} - ab\sigma A_{s,t} + ab\sigma^2 L_{s,t}^{(1)} + a^2 b\sigma L_{s,t}^{(2)},$$

$$z(0) = Y_s.$$

where $h := t - s$ denotes the step size and the random variables $A_{s,t}, L_{s,t}^{(1)}, L_{s,t}^{(2)}$ are given by

$$A_{s,t} := \int_s^t W_{s,r}\, \mathrm{d}r - \frac{1}{2}hW_{s,t},$$

$$L_{s,t}^{(1)} := \int_s^t \int_s^r W_{s,v} \circ \mathrm{d}W_v\, \mathrm{d}r - \frac{1}{2}W_{s,t}A_{s,t} - \frac{1}{6}hW_{s,t}^2,$$

$$L_{s,t}^{(2)} := \int_s^t \int_s^r W_{s,v}\, \mathrm{d}v\, \mathrm{d}r - \frac{1}{2}hA_{s,t} - \frac{1}{6}h^2 W_{s,t}.$$

In Foster et al. (2020), the depth-3 log-signature of $X = \{(t, W_t)\}_{t \geq 0}$ was approximated so that the above log-ODE method became practical and this numerical scheme exhibited state-of-the-art convergence rates. For example, the approximation error produced by 25 steps of the high order log-ODE method was similar to the error of the "increment only" log-ODE method with 1000 steps.

## B  CONVERGENCE OF THE LOG-ODE METHOD FOR ROUGH DIFFERENTIAL EQUATIONS

In this section, we shall present "rough path" error estimates for the log-ODE method. In addition, we will discuss the case when the vector fields governing the rough differential equation are linear. We begin by stating the main result of Boutaib et al. (2014) which quantifies the approximation error of the log-ODE method in terms of the regularity of the systems vector field $f$ and control path $X$. Since this section uses a number of technical definitions from rough path theory, we recommend Lyons et al. (2007) as an introduction to the subject.

For $T > 0$, we will use the notation $\triangle_T := \{(s, t) \in [0, T]^2 : s < t\}$ to denote a rescaled 2-simplex.

**Theorem B.1 (Lemma 15 in Boutaib et al. (2014))** *Consider the rough differential equation*

$$\mathrm{d}Y_t = f(Y_t)\, \mathrm{d}X_t, \tag{24}$$
$$Y_0 = \xi,$$

*where we make the following assumptions:*

- $X$ *is a* geometric $p$-rough path *in* $\mathbb{R}^d$, *that is* $X : \triangle_T \to T^{\lfloor p \rfloor}(\mathbb{R}^d)$ *is a continuous path in the tensor algebra* $T^{\lfloor p \rfloor}(\mathbb{R}^d) := \mathbb{R} \oplus \mathbb{R}^d \oplus (\mathbb{R}^d)^{\otimes 2} \oplus \cdots \oplus (\mathbb{R}^d)^{\otimes \lfloor p \rfloor}$ *with increments*

$$X_{s,t} = \left(1, X_{s,t}^{(1)}, X_{s,t}^{(2)}, \cdots, X_{s,t}^{(\lfloor p \rfloor)}\right), \tag{25}$$
$$X_{s,t}^{(k)} := \pi_k(X_{s,t}),$$

*where* $\pi_k : T^{\lfloor p \rfloor}(\mathbb{R}^d) \to (\mathbb{R}^d)^{\otimes k}$ *is the projection map onto* $(\mathbb{R}^d)^{\otimes k}$, *such that there exists a sequence of continuous finite variation paths* $x_n : [0, T] \to \mathbb{R}^d$ *whose truncated signatures converge to* $X$ *in the* $p$-variation metric*:*

$$d_p\left(S^{\lfloor p \rfloor}(x_n), X\right) \to 0, \tag{26}$$

*as* $n \to \infty$, *where the $p$-variation between two continuous paths* $Z^1$ *and* $Z^2$ *in* $T^{\lfloor p \rfloor}(\mathbb{R}^d)$ *is*

$$d_p(Z^1, Z^2) := \max_{1 \leq k \leq \lfloor p \rfloor} \sup_{\mathcal{D}} \left(\sum_{t_i \in \mathcal{D}} \left\|\pi_k(Z_{t_i, t_{i+1}}^1) - \pi_k(Z_{t_i, t_{i+1}}^2)\right\|^{\frac{p}{k}}\right)^{\frac{k}{p}}, \tag{27}$$

*where the supremum is taken over all partitions $\mathcal{D}$ of $[0, T]$ and the norms $\|\cdot\|$ must satisfy (up to some constant)*

$$\|a \otimes b\| \leq \|a\|\|b\|,$$

*for $a \in (\mathbb{R}^d)^{\otimes n}$ and $b \in (\mathbb{R}^d)^{\otimes m}$. For example, we can take $\|\cdot\|$ to be the projective or injective tensor norms (see Propositions 2.1 and 3.1 in Ryan (2002)).*

- *The solution $Y$ and its initial value $\xi$ both take their values in $\mathbb{R}^n$.*

- *The collection of vector fields $\{f_1, \cdots, f_d\}$ on $\mathbb{R}^n$ are denoted by $f : \mathbb{R}^n \to L(\mathbb{R}^n, \mathbb{R}^d)$, where $L(\mathbb{R}^n, \mathbb{R}^d)$ is the space of linear maps from $\mathbb{R}^n$ to $\mathbb{R}^d$. We will assume that $f$ has $\mathrm{Lip}(\boldsymbol{\gamma})$ regularity with $\gamma > p$. That is, $f$ it is bounded with $\lfloor\gamma\rfloor$ bounded derivatives, the last being Hölder continuous with exponent $(\gamma - \lfloor\gamma\rfloor)$. Hence the following norm is finite:*

$$\|f\|_{\mathrm{Lip}(\gamma)} := \max_{0 \leq k \leq \lfloor\gamma\rfloor} \left\|D^k f\right\|_\infty \vee \left\|D^{\lfloor\gamma\rfloor} f\right\|_{(\gamma-\lfloor\gamma\rfloor)-H\ddot{o}l}, \tag{28}$$

*where $D^k f$ is the $k$-th (Fréchet) derivative of $f$ and $\|\cdot\|_{\alpha\text{-}H\ddot{o}l}$ is the standard $\alpha$-Hölder norm with $\alpha \in (0, 1)$.*

- *The RDE (24) is defined in the Lyon's sense. Therefore by the Universal Limit Theorem (see Theorem 5.3 in Lyons et al. (2007)), there exists a unique solution $Y : [0, T] \to \mathbb{R}^n$.*

*We define the log-ODE for approximating the solution $Y$ over an interval $[s, t] \subset [0, T]$ as follows:*

1. *Compute the depth-$\lfloor\gamma\rfloor$ log-signature of the control path $X$ over $[s, t]$. That is, we obtain $\mathrm{LogSig}_{s,t}^{\lfloor\gamma\rfloor}(X) := \log_{\lfloor\gamma\rfloor}\left(S_{s,t}^{\lfloor\gamma\rfloor}(X)\right) \in T^{\lfloor\gamma\rfloor}(\mathbb{R}^d)$, where $\log_{\lfloor\gamma\rfloor}(\cdot)$ is defined by projecting the standard tensor logarithm map onto $\{a \in T^{\lfloor\gamma\rfloor}(\mathbb{R}^d) : \pi_0(a) > 0\}$.*

2. *Construct the following (well-posed) ODE on the interval $[0, 1]$,*

$$\frac{dz^{s,t}}{du} = F(z^{s,t}), \tag{29}$$
$$z_0^{s,t} = Y_s,$$

*where the vector field $F : \mathbb{R}^n \to \mathbb{R}^n$ is defined from the log-signature as*

$$F(z) := \sum_{k=1}^{\lfloor\gamma\rfloor} f^{\circ k}(z) \pi_k\left(\mathrm{LogSig}_{s,t}^{\lfloor\gamma\rfloor}(X)\right). \tag{30}$$

*Recall that $f^{\circ k} : \mathbb{R}^n \to L((\mathbb{R}^d)^{\otimes k}, \mathbb{R}^n)$ was defined previously in Definition A.6.*

*Then we can approximate $Y_t$ using the $u = 1$ solution of (29). Moreover, there exists a universal constant $C_{p,\gamma}$ depending only on $p$ and $\gamma$ such that*

$$\left\|Y_t - z_1^{s,t}\right\| \leq C_{p,\gamma} \|f\|_{\mathrm{Lip}(\gamma)}^\gamma \|X\|_{p\text{-}var;[s,t]}^\gamma, \tag{31}$$

*where $\|\cdot\|_{p\text{-}var;[s,t]}$ is the $p$-variation norm defined for paths in $T^{\lfloor p\rfloor}(\mathbb{R}^d)$ by*

$$\|X\|_{p\text{-}var;[s,t]} := \max_{1 \leq k \leq \lfloor p\rfloor} \sup_{\mathcal{D}} \left(\sum_{t_i \in \mathcal{D}} \left\|X_{t_i,t_{i+1}}^k\right\|^{\frac{p}{k}}\right)^{\frac{k}{p}}, \tag{32}$$

*with the supremum taken over all partitions $\mathcal{D}$ of $[s, t]$.*

**Remark B.2** *If the vector fields $\{f_1, \cdots, f_d\}$ are linear, then it immediately follows that $F$ is linear.*

Although the above theorem requires some sophisticated theory, it has a simple conclusion - namely that log-ODEs can approximate controlled differential equations. That said, the estimate (31) does not directly apply when the vector fields $\{f_i\}$ are linear as they would be unbounded. Fortunately, it is well known that linear RDEs are well posed and the growth of their solutions can be estimated.

**Theorem B.3 (Theorem 10.57 in Friz & Victoir (2010))** *Consider the linear RDE on $[0, T]$*

$$\mathrm{d}Y_t = f(Y_t)\,\mathrm{d}X_t,$$
$$Y_0 = \xi,$$

*where $X$ is a geometric p-rough path in $\mathbb{R}^d$, $\xi \in \mathbb{R}^n$ and the vector fields $\{f_i\}_{1 \le i \le d}$ take the form $f_i(y) = A_i y + B$ where $\{A_i\}$ and $\{B_i\}$ are $n \times n$ matrices. Let $K$ denote an upper bound on $\max_i(\|A_i\| + \|B_i\|)$. Then a unique solution $Y : [0, T] \to \mathbb{R}^n$ exists. Moreover, it is bounded and there exists a constant $C_p$ depending only on $p$ such that*

$$\|Y_t - Y_s\| \le C_p\big(1 + \|\xi\|\big)K\|X\|_{p\text{-}var;[s,t]} \exp\Big(C_p K^p \|X\|_{p\text{-}var;[s,t]}^p\Big), \tag{33}$$

*for all $0 \le s \le t \le T$.*

When the vector fields of the RDE (24) are linear, then the log-ODE (29) also becomes linear. Therefore, the log-ODE solution exists and is explicitly given as the exponential of the matrix $F$.

**Theorem B.4** *Consider the same linear RDE on $[0, T]$ as in Theorem B.3,*

$$\mathrm{d}Y_t = f(Y_t)\,\mathrm{d}X_t,$$
$$Y_0 = \xi.$$

*Then the log-ODE vector field $F$ given by (30) is linear and the solution of the associated ODE (29) exists and satisfies*

$$\|z_u^{s,t}\| \le \|Y_s\| \exp\bigg( \sum_{m=1}^{\lfloor \gamma \rfloor} K^m \Big\|\pi_m\Big(\mathrm{LogSig}_{s,t}^{\lfloor \gamma \rfloor}(X)\Big)\Big\| \bigg), \tag{34}$$

*for $u \in [0, 1]$ and all $0 \le s \le t \le T$.*

**Proof B.5** *Since $F$ is a linear vector field on $\mathbb{R}^n$, we can view it as an $n \times n$ matrix and so for $u \in [0, 1]$,*

$$z_u^{s,t} = \exp(uF)z_0^{s,t},$$

*where $\exp$ denotes the matrix exponential. The result now follows by the standard estimate $\|\exp(F)\| \le \exp(\|F\|)$.*

**Remark B.6** *Due to the boundedness of linear RDEs (33) and log-ODEs (34), the arguments that established Theorem B.1 will hold in the linear setting as $\|f\|_{\mathrm{Lip}(\gamma)}$ would be finite when defined on the domains that the solutions $Y$ and $z$ lie in.*

Given the local error estimate (31) for the log-ODE method, we can now consider the approximation error that is exhibited by a log-ODE numerical solution to the RDE (24). Thankfully, the analysis required to derive such global error estimates was developed by Greg Gyurkó in his PhD thesis. Thus the following result is a straightforward application of Theorem 3.2.1 from Gyurkó (2008).

**Theorem B.7** *Let $X$, $f$ and $Y$ satisfy the assumptions given by Theorem B.1 and suppose that $\{0 = t_0 < t_1 < \cdots < t_N = T\}$ is a partition of $[0, T]$ with $\max_k \|X\|_{p\text{-}var;[t_k, t_{k+1}]}$ sufficiently small. We can construct a numerical solution $\{Y_k^{\log}\}_{0 \le k \le N}$ of (24) by setting $Y_0^{\log} := Y_0$ and for each $k \in \{0, 1, \cdots, N-1\}$, defining $Y_{k+1}^{\log}$ to be the solution at $u = 1$ of the following ODE:*

$$\frac{\mathrm{d}z^{t_k, t_{k+1}}}{\mathrm{d}u} := F\big(z^{t_k, t_{k+1}}\big), \tag{35}$$
$$z_0^{t_k, t_{k+1}} := Y_k^{\log},$$

*where the vector field $F$ is constructed from the log-signature of $X$ over the interval $[t_k, t_{k+1}]$ according to (30). Then there exists a constant $C$ depending only on $p$, $\gamma$ and $\|f\|_{\mathrm{Lip}(\gamma)}$ such that*

$$\big\|Y_{t_k} - Y_k^{\log}\big\| \le C \sum_{i=0}^{k-1} \|X\|_{p\text{-}var;[t_i, t_{i+1}]}^{\gamma}, \tag{36}$$

*for $0 \le k \le N$.*

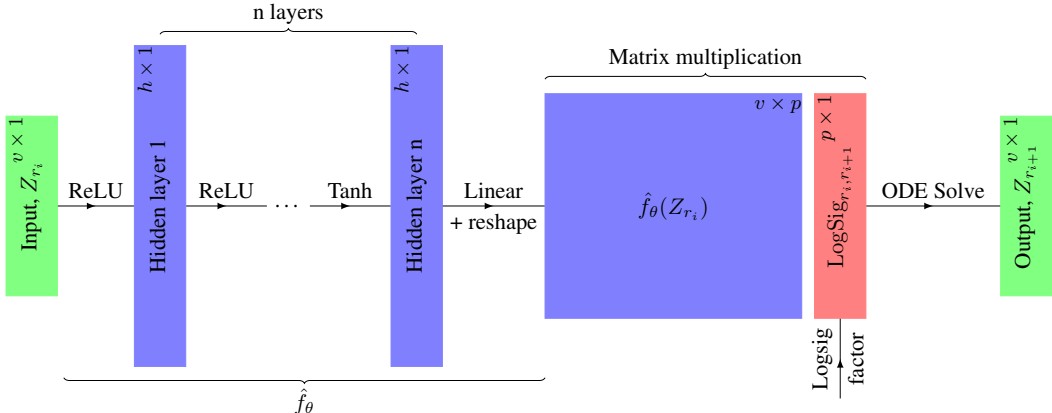

Figure 6: Overview of the hidden state update network structure. We give the dimensions at each layer in the top right hand corner of each box.

**Remark B.8** *The above error estimate also holds when the vector field $f$ is linear (by Remark B.6)).*

Since $\lfloor \gamma \rfloor$ is the truncation depth of the log-signatures used to construct each log-ODE vector field, we see that high convergence rates can be achieved through using more terms in each log-signature. It is also unsurprising that the error estimate (36) increases with the "roughness" of the control path. So just as in our experiments, we see that the performance of the log-ODE method can be improved by choosing an appropriate step size and depth of log-signature.

## C  EXPERIMENTAL DETAILS

**Code**  The code to reproduce the experiments is available at [redacted; see supplementary material]

**Data splits**  Each dataset was split into a training, validation, and testing dataset with relative sizes 70%/15%/15%.

**Normalisation**  The training splits of each dataset were normalised to zero mean and unit variance. The statistics from the training set were then used to normalise the validation and testing datasets.

**Architecture**  We give a graphical description of the architecture used for updating the Neural CDE hidden state in figure 6. The input is first run through a multilayer perceptron with $n$ layers of size $h$, with with $n, h$ being hyperparameters. ReLU nonlinearities are used at each layer except the final one, where we instead use a tanh nonlinearity. The goal of this is to help prevent term blow-up over the long sequences.

Note that this is a small inconsistency between this work and the original model proposed in Kidger et al. (2020). Here, we applied the tanh function as the final hidden layer nonlinearity, whilst in the original paper the tanh nonlinearity is applied after the final linear map. Both methods are used to constrain the rate of change of the hidden state; we do not know of a reason to prefer one over the other.

Note that the final linear layer in the multilayer perceptron is reshaped to produce a matrix-valued output, of shape $v \times p$. (As $\widehat{f_\theta}$ is matrix-valued.) A matrix-vector multiplication with the log-signature then produces the vector field for the ODE solver.

**ODE Solver**  All problems used the 'rk4' solver as implemented by `torchdiffeq` (Chen, 2018) version 0.0.1.

**Computing infrastructure**    All EigenWorms experiments were run on a computer equipped with three GeForce RTX 2080 Ti's. All BIDMC experiments were run on a computed with two GeForce RTX 2080 Ti's and two Quadro GP100's.

**Optimiser**    All experiments used the Adam optimiser. The learning rate was initialised at 0.032 divided by batch size. The batch size used was 1024 for EigenWorms and 512 for the BIDMC problems. If the validation loss failed to decrease after 15 epochs the learning rate was reduced by a factor of 10. If the validation loss did not decrease after 60 epochs, training was terminated and the model was rolled back to the point at which it achieved the lowest loss on the validation set.

**Hyperparameter selection**    Hyperparameters were selected to optimise the score of the $NCDE_1$ model on the validation set. For each dataset the search was performed with a step size that meant the total number of hidden state updates was equal to 500, as this represented a good balance between length and speed that allowed us to complete the search in a reasonable time-frame. In particular, this was short enough that we could train using the non-adjoint training method which helped to speed this section up. The hyperparameters that were considered were:

- Hidden dimension: [16, 32, 64] - The dimension of the hidden state $Z_t$.
- Number of layers: [2, 3, 4] - The number of hidden state layers.
- Hidden hidden multiplier: [1, 2, 3] - Multiplication factor for the hidden hidden state, this being the 'Hidden layer $k$' in figure 6. The dimension of each of these 'hidden hidden' layers with be this value multiplied by 'Hidden dimension'.

We ran each of these 27 total combinations for every dataset and the parameters that corresponded were used as the parameters when training over the full depth and step grid. The full results from the hyperparameter search are listed in tables (3, 4) with bolded values to show which values were eventually selected.

## D    EXPERIMENTAL RESULTS

Here we include the full breakdown of all experimental results. Tables 5 and 6 include all results from the EigenWorms and BIDMC datasets respectively.

| Validation accuracy | Hidden dim | Num layers | Hidden hidden multiplier | Total params |
|:---:|:---:|:---:|:---:|:---:|
| 33.3 | 16 | 2 | 3 | 5509 |
| 43.6 | 16 | 2 | 2 | 5509 |
| 56.4 | 16 | 2 | 1 | 4453 |
| 64.1 | 16 | 3 | 2 | 8869 |
| 38.5 | 16 | 3 | 3 | 8869 |
| 51.3 | 16 | 3 | 1 | 6517 |
| 82.1 | 16 | 4 | 2 | 12741 |
| 35.9 | 16 | 4 | 3 | 12741 |
| 53.8 | 16 | 4 | 1 | 8581 |
| 35.9 | 32 | 2 | 3 | 21253 |
| 74.4 | 32 | 2 | 2 | 21253 |
| 43.6 | 32 | 2 | 1 | 17093 |
| 53.8 | 32 | 3 | 3 | 34629 |
| **87.2** | **32** | **3** | **2** | **34629** |
| 64.1 | 32 | 3 | 1 | 25317 |
| 35.9 | 32 | 4 | 3 | 50053 |
| 71.8 | 32 | 4 | 1 | 33541 |
| 79.5 | 32 | 4 | 2 | 50053 |
| 41.0 | 64 | 2 | 3 | 83461 |
| 64.1 | 64 | 2 | 2 | 83461 |
| 48.7 | 64 | 3 | 3 | 136837 |
| 59.0 | 64 | 3 | 2 | 136837 |
| 51.3 | 64 | 2 | 1 | 66949 |
| 56.4 | 64 | 4 | 2 | 198405 |
| 64.1 | 64 | 4 | 3 | 198405 |
| 64.1 | 64 | 3 | 1 | 99781 |
| 51.3 | 64 | 4 | 1 | 132613 |

Table 3: Hyperparamter selection results for the EigenWorms dataset. The blue values denote the selected hyperparameters.

| Validation loss | | | Hidden dim | Num layers | Hidden hidden multiplier | Total params |
|------|------|------|------------|------------|--------------------------|--------------|
| RR | HR | SpO2 | | | | |
| 1.72 | 6.10 | 2.07 | 16 | 2 | 1 | 2209 |
| 1.57 | 5.58 | 1.97 | 16 | 2 | 2 | 3265 |
| 1.55 | 6.10 | 1.33 | 16 | 2 | 3 | 3265 |
| 1.80 | 5.16 | 2.05 | 16 | 3 | 1 | 3249 |
| 1.61 | 5.22 | 1.62 | 16 | 3 | 2 | 5601 |
| 1.56 | 3.34 | 1.18 | 16 | 3 | 3 | 5601 |
| 1.57 | 3.86 | 1.97 | 16 | 4 | 1 | 4289 |
| 1.45 | 3.54 | 1.25 | 16 | 4 | 2 | 8449 |
| 1.54 | 3.93 | 1.09 | 16 | 4 | 3 | 8449 |
| 1.56 | 6.81 | 1.87 | 32 | 2 | 1 | 8513 |
| 1.42 | 3.11 | 1.43 | 32 | 2 | 2 | 12673 |
| 1.54 | 3.60 | 1.11 | 32 | 2 | 3 | 12673 |
| 1.54 | 3.52 | 1.57 | 32 | 3 | 1 | 12641 |
| 1.39 | 2.96 | 1.03 | 32 | 3 | 2 | 21953 |
| 1.47 | 2.95 | 1.05 | 32 | 3 | 3 | 21953 |
| 1.55 | 3.00 | 2.00 | 32 | 4 | 1 | 16769 |
| 1.38 | 3.20 | 1.07 | 32 | 4 | 2 | 33281 |
| 1.43 | 2.58 | 1.01 | 32 | 4 | 3 | 33281 |
| 1.51 | 3.21 | 1.10 | 64 | 2 | 1 | 33409 |
| 1.43 | **2.22** | 1.00 | **64** | **2** | **2** | **49921** |
| 1.51 | 3.34 | 0.94 | 64 | 2 | 3 | 49921 |
| 1.55 | 3.24 | 2.09 | 64 | 3 | 1 | 49857 |
| 1.32 | 2.53 | 0.88 | 64 | 3 | 2 | 86913 |
| **1.25** | 2.57 | **0.73** | **64** | **3** | **3** | **86913** |
| 1.43 | 5.78 | 1.43 | 64 | 4 | 1 | 66305 |
| 1.28 | 2.26 | 0.93 | 64 | 4 | 2 | 132097 |
| 1.32 | 2.46 | 1.15 | 64 | 4 | 3 | 132097 |

Table 4: Hyperparameter selection results for each problem of the BIDMC dataset. The bold values denote the selected hyperparameters for each vitals sign problem. Note that RR and SpO2 had the same parameters selected, hence why only two lines are given in bold.

| Model | Step | Test Accuracy | Time (Hrs) | Memory (Mb) |
|---|---|---|---|---|
| NCDE$_1$ | 1 | $62.4 \pm 12.1$ | 22.0 | 176.5 |
|  | 2 | $69.2 \pm 4.4$ | 14.6 | 90.6 |
|  | 4 | $66.7 \pm 11.8$ | 5.5 | 46.6 |
|  | 6 | $65.8 \pm 12.9$ | 2.6 | 31.5 |
|  | 8 | $64.1 \pm 13.3$ | 3.1 | 24.3 |
|  | 16 | $64.1 \pm 16.8$ | 1.5 | 13.4 |
|  | 32 | $64.1 \pm 14.3$ | 0.5 | 8.0 |
|  | 64 | $56.4 \pm 6.8$ | 0.4 | 5.2 |
|  | 128 | $48.7 \pm 2.6$ | 0.1 | 3.9 |
|  | 256 | $42.7 \pm 3.0$ | 0.1 | 3.2 |
|  | 512 | $44.4 \pm 5.3$ | 0.0 | 2.9 |
|  | 1024 | $41.9 \pm 14.6$ | 0.0 | 2.7 |
|  | 2048 | $38.5 \pm 5.1$ | 0.0 | 2.6 |
| NCDE$_2$ | 2 | $\mathbf{76.1 \pm 13.2}$ | 9.8 | 354.3 |
|  | 4 | $\mathbf{83.8 \pm 3.0}$ | 2.4 | 180.0 |
|  | 6 | $\mathbf{76.9 \pm 6.8}$ | 2.0 | 82.2 |
|  | 8 | $\mathbf{77.8 \pm 5.9}$ | 2.1 | 94.2 |
|  | 16 | $\mathbf{78.6 \pm 3.9}$ | 1.3 | 50.2 |
|  | 32 | $67.5 \pm 12.1$ | 0.7 | 28.1 |
|  | 64 | $73.5 \pm 7.8$ | 0.4 | 17.2 |
|  | 128 | $\mathbf{76.1 \pm 5.9}$ | 0.2 | 7.8 |
|  | 256 | $\mathbf{72.6 \pm 12.1}$ | 0.1 | 8.9 |
|  | 512 | $\mathbf{69.2 \pm 11.8}$ | 0.0 | 7.6 |
|  | 1024 | $\mathbf{65.0 \pm 7.4}$ | 0.0 | 6.9 |
|  | 2048 | $\mathbf{67.5 \pm 3.9}$ | 0.0 | 6.5 |
| NCDE$_3$ | 2 | $66.7 \pm 4.4$ | 7.4 | 1766.2 |
|  | 4 | $76.9 \pm 9.2$ | 2.8 | 856.8 |
|  | 6 | $70.9 \pm 1.5$ | 1.4 | 606.1 |
|  | 8 | $70.1 \pm 6.5$ | 1.3 | 460.7 |
|  | 16 | $73.5 \pm 3.0$ | 1.4 | 243.7 |
|  | 32 | $\mathbf{75.2 \pm 3.0}$ | 0.6 | 134.7 |
|  | 64 | $\mathbf{74.4 \pm 11.8}$ | 0.3 | 81.0 |
|  | 128 | $68.4 \pm 8.2$ | 0.1 | 53.3 |
|  | 256 | $60.7 \pm 8.2$ | 0.1 | 40.2 |
|  | 512 | $62.4 \pm 10.4$ | 0.0 | 33.1 |
|  | 1024 | $59.8 \pm 3.9$ | 0.0 | 29.6 |
|  | 2048 | $61.5 \pm 4.4$ | 0.0 | 27.7 |

Table 5: Mean and standard deviation of test set accuracy (in %) over three repeats, as well as memory usage and training time, on the EigenWorms dataset for depths 1–3 and a small selection of step sizes. The bold values denote that the model was the top performer for that step size.

| Depth | Step | $L^2$ | | | Time (H) | | | Memory (Mb) |
|---|---|---|---|---|---|---|---|---|
| | | RR | HR | $SpO_2$ | RR | HR | $SpO_2$ | |
| NCDE$_1$ | 1 | $2.79 \pm 0.04$ | $9.82 \pm 0.34$ | $2.83 \pm 0.27$ | 23.8 | 22.1 | 28.1 | 56.5 |
| | 2 | $2.87 \pm 0.03$ | $11.69 \pm 0.38$ | $\mathbf{3.36 \pm 0.2}$ | 19.3 | 9.6 | 8.8 | 32.6 |
| | 4 | $\mathbf{2.92 \pm 0.08}$ | $11.15 \pm 0.49$ | $3.69 \pm 0.06$ | 5.3 | 5.7 | 3.2 | 20.2 |
| | 8 | $2.8 \pm 0.06$ | $10.72 \pm 0.24$ | $3.43 \pm 0.17$ | 3.0 | 2.6 | 4.8 | 14.3 |
| | 16 | $2.22 \pm 0.07$ | $7.98 \pm 0.61$ | $2.9 \pm 0.11$ | 1.7 | 1.4 | 1.8 | 11.8 |
| | 32 | $2.53 \pm 0.23$ | $12.23 \pm 0.43$ | $2.68 \pm 0.12$ | 1.9 | 0.9 | 2.2 | 9.8 |
| | 64 | $2.63 \pm 0.11$ | $12.02 \pm 0.09$ | $2.88 \pm 0.06$ | 0.2 | 0.3 | 0.4 | 9.1 |
| | 128 | $2.64 \pm 0.18$ | $11.98 \pm 0.37$ | $2.86 \pm 0.04$ | 0.2 | 0.2 | 0.3 | 8.7 |
| | 256 | $2.53 \pm 0.04$ | $12.29 \pm 0.1$ | $3.08 \pm 0.1$ | 0.1 | 0.1 | 0.1 | 8.3 |
| | 512 | $2.53 \pm 0.03$ | $12.22 \pm 0.11$ | $2.98 \pm 0.04$ | 0.1 | 0.0 | 0.1 | 8.4 |
| | 1024 | $2.67 \pm 0.12$ | $11.55 \pm 0.03$ | $2.91 \pm 0.12$ | 0.1 | 0.1 | 0.1 | 8.4 |
| | 2048 | $2.48 \pm 0.03$ | $12.03 \pm 0.2$ | $3.25 \pm 0.01$ | 0.0 | 0.1 | 0.0 | 8.2 |
| NCDE$_2$ | 2 | $2.91 \pm 0.1$ | $11.11 \pm 0.23$ | $3.89 \pm 0.44$ | 12.7 | 9.3 | 8.2 | 58.3 |
| | 4 | $\mathbf{2.92 \pm 0.04}$ | $11.14 \pm 0.2$ | $4.23 \pm 0.57$ | 18.1 | 5.0 | 3.4 | 34.0 |
| | 8 | $2.63 \pm 0.12$ | $8.63 \pm 0.24$ | $2.88 \pm 0.15$ | 2.1 | 3.4 | 3.3 | 21.8 |
| | 16 | $1.8 \pm 0.07$ | $5.73 \pm 0.45$ | $1.98 \pm 0.21$ | 2.2 | 1.4 | 2.5 | 16.0 |
| | 32 | $1.9 \pm 0.02$ | $7.9 \pm 1.0$ | $1.69 \pm 0.2$ | 1.2 | 1.1 | 2.0 | 13.1 |
| | 64 | $1.89 \pm 0.04$ | $5.54 \pm 0.45$ | $2.04 \pm 0.07$ | 0.3 | 0.3 | 1.7 | 11.6 |
| | 128 | $1.86 \pm 0.03$ | $6.77 \pm 0.42$ | $1.95 \pm 0.18$ | 0.3 | 0.4 | 0.7 | 10.9 |
| | 256 | $1.86 \pm 0.09$ | $5.64 \pm 0.19$ | $2.1 \pm 0.19$ | 0.1 | 0.1 | 0.5 | 10.5 |
| | 512 | $1.81 \pm 0.02$ | $5.05 \pm 0.23$ | $2.17 \pm 0.18$ | 0.1 | 0.2 | 0.4 | 10.3 |
| | 1024 | $1.93 \pm 0.11$ | $6.0 \pm 0.19$ | $2.41 \pm 0.07$ | 0.1 | 0.1 | 0.2 | 10.2 |
| | 2048 | $\mathbf{2.03 \pm 0.03}$ | $\mathbf{7.7 \pm 1.46}$ | $2.55 \pm 0.03$ | 0.1 | 0.1 | 0.1 | 10.2 |
| NCDE$_3$ | 2 | $\mathbf{2.82 \pm 0.08}$ | $\mathbf{11.01 \pm 0.28}$ | $4.1 \pm 0.72$ | 8.8 | 9.4 | 6.9 | 125.2 |
| | 4 | $2.97 \pm 0.23$ | $\mathbf{10.13 \pm 0.62}$ | $\mathbf{3.56 \pm 0.44}$ | 3.2 | 4.1 | 2.6 | 71.6 |
| | 8 | $\mathbf{2.42 \pm 0.19}$ | $\mathbf{7.67 \pm 0.4}$ | $\mathbf{2.55 \pm 0.13}$ | 2.9 | 3.2 | 3.1 | 43.3 |
| | 16 | $\mathbf{1.74 \pm 0.05}$ | $\mathbf{4.11 \pm 0.61}$ | $\mathbf{1.4 \pm 0.06}$ | 1.4 | 1.4 | 6.5 | 29.1 |
| | 32 | $\mathbf{1.67 \pm 0.01}$ | $\mathbf{4.5 \pm 0.7}$ | $\mathbf{1.61 \pm 0.05}$ | 1.3 | 1.8 | 7.3 | 20.5 |
| | 64 | $\mathbf{1.53 \pm 0.08}$ | $\mathbf{3.05 \pm 0.36}$ | $\mathbf{1.48 \pm 0.14}$ | 0.4 | 1.9 | 3.3 | 17.9 |
| | 128 | $\mathbf{1.51 \pm 0.08}$ | $\mathbf{2.97 \pm 0.45}$ | $\mathbf{1.37 \pm 0.22}$ | 0.5 | 1.7 | 1.7 | 17.3 |
| | 256 | $\mathbf{1.51 \pm 0.06}$ | $\mathbf{3.4 \pm 0.74}$ | $\mathbf{1.47 \pm 0.07}$ | 0.3 | 0.7 | 0.6 | 16.6 |
| | 512 | $\mathbf{1.49 \pm 0.08}$ | $\mathbf{3.46 \pm 0.13}$ | $\mathbf{1.29 \pm 0.15}$ | 0.3 | 0.4 | 0.4 | 15.4 |
| | 1024 | $\mathbf{1.83 \pm 0.33}$ | $\mathbf{5.58 \pm 2.5}$ | $\mathbf{1.72 \pm 0.31}$ | 0.2 | 0.1 | 0.1 | 15.7 |
| | 2048 | $2.31 \pm 0.27$ | $9.77 \pm 1.53$ | $\mathbf{2.45 \pm 0.18}$ | 0.1 | 0.1 | 0.1 | 15.6 |

Table 6: Mean and standard deviation of the $L^2$ losses on the test set for each of the vitals signs prediction tasks (RR, HR, $SpO_2$) on the BIDMC dataset, across three repeats. Only mean times are shown for space. The memory usage is given as the mean over all three of the tasks as it was approximately the same for any task for a given depth and step. The bold values denote the algorithm with the lowest test set loss for a fixed step size for each task.

