# OpenReview forum: "Neural CDEs for Long Time Series via the Log-ODE Method"
_ICLR.cc/2021/Conference — Reject_

### Official Review · AnonReviewer5 · 2020-10-27
**Seems novel and motivated, but should be more approachable.**

**Rating:** 6
**Confidence:** 4

**Review:**

### **Summary and Contributions of Paper**
This paper proposes a new method for computing Neural CDEs via the signature transform, which transforms a path integral into log signatures, i.e. a collection of iterated integrals. Then standard ODE tools are applied to each piecewise log signature.

### **Strengths**
- The writing quality is rigorous.
- The approach seems motivated and based on a clever mathematical trick via the signature transform.
- Experiments are convincing and sound.
- Appendix provides proof of the approximation properties of the clipped-term signature transform (which originally requires infinite basis for exact approximation)

### **Weaknesses**
- The signature transform seems somewhat esoteric and nonstandard for readers without specific knowledge in this field. It would be very good if the authors could give more intuitive/pictorial views of this transform (I needed to read online surveys multiple times to understand the intuition behind this). For instance, I read this in detail: https://arxiv.org/pdf/1905.08494.pdf (NeurIPS 2019), which provides a much cleaner explanation of the signature transform, but also demonstrates that an entire paper is needed to simply explain the method.
- While the authors claim that there is an ease of implementation via pre-existing tools, the larger bottleneck seems to be actually understanding the method itself (which seems to also be a function of how the paper treats this material). While I have no doubt that this work would be great for a very mathematically minded community, I am unsure of its merits for the ICLR conference community. I think the authors should provide more high level overview of the signature transform, and keep the strict math in the appendix.

I am not an expert on these types of methods, so my confidence will not be as high, but I believe that this paper contributes via its insight with the signature transform, and thus my rating is marginally above the acceptance threshold.

If the authors could perhaps make the work more approachable, I would be happy to raise my score.

---

> ### Author Response · Authors · 2020-11-12
> **Response**
>
> Thank you for your review.
>
> **On approachability**
> We have worked to simplify the presentation so as to make the work more approachable.
>
> Most importantly, we have used the additional space to introduce further intuition of these ideas. In particular we have:
>
> - added an introduction to the signature and log-signature section, giving a high-level overview before giving their mathematical definition;
> - added pictorial descriptions giving a geometric insight into the (log-)signature transform;
> - demonstrated with a Taylor expansion how signature terms naturally arise in the solution of CDEs
>
> Additionally:
>
> - We have replaced the definition of the signature transform with the simpler-to-understand definition previously used in the "Deep Signature Transforms" (NeurIPS 2019) paper, that the reviewer highlights;
> - The definition of $\beta(d, n)$ (the size of the logsignature) has been moved to the appendix;
> - Where possible, mathematical terms (in particular "Riemann--Stieltjes integrals"; "n-fold iterated integral";  "Lie brackets"; "Magnus expansion") have been removed, when the discussion could be rephrased without them;
> - The term "tree-like equivalance" has been replaced with a similar statement on translation invariance;
> - Holder continuity has been replaced with Lipschitz continuity (which is weaker but better-known).
>
> We completely recognise the difficulty of these concepts if the theory is unfamiliar. It is precisely because of this that the start of Section 2 ("Theory") begins with *"We begin with motivating theory, though we note that this section is not essential for using the method. Readers more interested in practical applications should feel free to skip to section 3",* and the rest of the paper is careful to be independent of this section. We think this represents the best possible compromise between readability and technicality, and have now added an additional sentence to help emphasise this.
>
> **On choice of venue**
> We have submitted to ICLR because our paper is about improving an already-existing machine learning technique.
>
> **Summary**
> We hope that these improvements are sufficient to address your concerns for an improved score. Please let us know if you have any additional questions or feedback, and we will be happy to address these as well.

---

> > ### Comment · AnonReviewer5 · 2020-11-23
> > **Paper is much cleaner**
> >
> > Thank you very much for the update to the paper - it reads *much much* cleaner now and I understand better the motivations and overall picture.
> >
> > I raise my confidence score to 4, but currently keep my rating as 6 - I do think that the paper has its merits and tends to be accepted, but I do not necessarily think that this is groundbreaking or very novel as it is mainly a data preprocessing technique applied to a specific set of problems (as noted by Reviewer 4).

---

### Official Review · AnonReviewer1 · 2020-10-28
**The training time of neural CDEs is reduced by replacing the original input time series with corresponding log-signature.**

**Rating:** 7
**Confidence:** 4

**Review:**

Summarizing the paper claims
------------------------------------------
The paper introduces an approach that allows training Neural Controlled Differential Equations (CDEs) for long time series. In contrast to the Neural ODE that is determined to its initial condition, Neural CDE produces a trajectory dependent on time-varying data. The authors propose to use log-signature as input instead of the original time series. Log-signature can be understood as a lossy representation for time series, which has a much smaller length and varies slower over the same time interval. Hence, larger steps may be used in the numerical solver, and that yields to the training speed-up.

In a little more detail, the log-signature is constructed as a sequence of statistics computed using the original time series. It is characterized by the depth N, where N states for the maximal length of paths used to calculate statistics. By applying log-signature transform, the solution of the CDE may be approximated by the solution of the ODE.

Strong points
-------------------
- The paper is clearly written.
- The experiments are conducted for four real-wold problems (worms classification, predicting a person's heart rate/respiratory rate/oxygen saturation).
- The authors provide an ablation study for the (log-signature length) / (number of channels in time series) trade-off,  investigate an influence of log-signature depth and solver's number of steps to the performance.
- The paper states the limitations of the proposed method.

Weak points
-----------------
The paper provides an overview of many RNN based models used for long time series; however, it doesn't compare with any of them. That would be interesting to see how good is a proposed method comparing to the RNN based one in terms of test accuracy/training time/memory usage.

Recommendation (accept or reject)
------------------------------------------------
I recommend accepting the paper. The paper provides a detailed theoretical formulation and demonstrates a significant training time speed-up for various real-world problems.

Update: The authors thoroughly addressed all the questions, the experiments demonstrate an improvement, the theory coincides with experiments. From my perspective, that would be useful for the neural ODE community to know more about the proposed log-signature-based technique. I increase the score.

Questions
--------------
- Which type of ODE solvers has been used for the experiments? Does the solver's order influence the neural CDE performance?
- What order is the value of  $\beta(v, N)$ in the experiments?
- In section 4.1, it is written that the naive subsampling achieves speed-up without performance improvements. Could you provide time and accuracy for these experiments?
-  What architecture is used for $\hat{f}$ neural network? How the choice of architecture affects performance?
- The standard deviation in Table 1 has quite large values. Haven't you tried to tune training hyperparameters to reduce it?
- What is the stopping criterion for the training? I'm curious why in Table 1, for the same number of solver's steps, the training time for $NCDE_2$ is longer than for  $NCDE_3$? (It seems that for the same number of training epochs, the time should be shorter because we do less preprocessing computations for log-transformation)

---

> ### Author Response · Authors · 2020-11-12
> **Response**
>
> Thank you for your review.
>
>  **Comparison to an RNN Baseline**
> RNN-based models do not fit in the memory of the GPU resources we have available. This is one of the main advantages of using differential equation models in the first place, for which adjoint backpropagation is available. (As per the first paragraph of Section 4.) Indeed, our focus is on ODE (CDE) modelling, for continuous time modelling, adjoint backpropagation, or the ability to easily handle irregular and partially observed time series. (In this case, adjoint backpropagation.)
>
> We have added additional commentary to the experimental section to help make this clear.
>
>  **Responses to Questions**
>
> - The solver used is a fourth-order Runge-Kutta with 3/8 rule ("rk4"). This is specified in the appendix. We did also investigate Euler and midpoint solvers, but found that they gave typically worse performance.
> - The order of $\beta(v, N)$ is to the best of our knowledge an open question! We have added a remark to this effect.
> - Naive subsampling is already included. This corresponds to an NCDE depth 1 model with step > 1 (top rows of results), since this is just the NCDE updated at t=t_0, t=t_0+step, t=t_0+2step, ... . We have added additional commentary to make this clear.
> - The architecture of $\widehat{f}$ is specified in the appendix, as a feedforward neural network with number of hidden layers in {1, 2, 3} and ReLU activation functions. The sizes of the network were optimised as hyperparameters.
> - On that note, we did indeed perform hyperparameter searching. The standard deviations reported are we then got.
> - Thank you for catching this - there was indeed an early stopping criterion if the training loss failed to improve over 60 epochs, which explains this difference. You are correct in assuming over the same number of training epochs the training time should be longer. We have added an additional comment explaining this.
>
>  **Summary**
> We hope this addresses all of the reviewer's concerns. If the reviewer has any further questions by which our paper and their score may be improved, then we would be happy to address these as well.

---

> > ### Comment · AnonReviewer1 · 2020-11-21
> > **A minor suggestion**
> >
> > Thank you for addressing all the questions.
> >
> > A minor suggestion: I recommend explicitly provide values for n and m in the experimental section (Section 4) for easier matching with the method description from Section 3.
> >
> > P.S.: an added geometric interpretation of log-signatures (Figure 2) is indeed helpful for understanding.

---

### Official Review · AnonReviewer4 · 2020-10-29
**Review of the "CDEs & the Log-ODE Method" for ICLR 2021**

**Rating:** 5
**Confidence:** 5

**Review:**

**Summary.** The authors describe how to apply a log signature to temporal datasets. This operation reduces dimensionality along the time axis at the price of adding some dimensionality to the spatial dimension. Then they train a neural controlled differential equation (Neural CDE) on the transformed dataset and show that their model learns more quickly and achieves better test generalization. They report results on two real-world datasets (EigenWorms and the TSR vitals dataset).

**Strong points.** This paper is technically sound and the method shows clear improvement over “no preprocessing.”

**Weak points.** The authors are proposing a simple method of reparameterizing time series data so that information is transferred from the time dimension to the space dimension. Strangely, the title “Neural CDEs for Long Time Series via the Log-ODE Method” implies that they are proposing a new model. They are not. Rather, they are proposing a data preprocessing technique that they apply to a dataset _before_ they train an ML model on it. It’s also worth noting that this feature engineering technique is already being used on time series data (eg Liao, 2019, “Learning Stochastic…”). The only difference is that here the authors are using a continuous-time analogue of an RNN. But since this feature engineering trick is applied to the dataset independently of what model is used, I’m not sure what new scientific insights are to be gained. This work seems very similar to Liao et al (2019); compare Figure 1 in the two papers, for example. To the authors: am I missing a critical new contribution? In what ways is the “log signature” feature engineering trick significantly different in the context of CDEs compared to RNNs?

Another issue with this paper is that it presents the theory in an unnecessarily mathematical manner. Some math is useful. But whenever the authors introduce terms that an ML audience is not familiar with, they should offer a sentence of intuition regarding what that term means. From my notes on this paper, here are phrases that I found confusing and were not accompanied by any explanation or intuition: “Riemann–Stieltjes integral,” “n-fold iterated integral,” “tree-like equivalence,” “Holder continuous paths,” “Mobius function,” “Lie brackets whose foliage is a Lindon word,” and “Magnus expansion.” Again, it’s ok to introduce these concepts, but 1) you should give some reasonable intuition for what they are and why they are relevant and 2) they should be directly relevant to the main contributions of the paper.

For other reviewers/readers who want an intuitive introduction to log signatures, I thought that [this README](https://github.com/kormilitzin/the-signature-method-in-machine-learning) did a very nice job. To the authors: are there ways that we can simplify the theory section so that it remains technically correct while also being readable and accessible? Are there specific reasons that you chose to introduce the "log signature” operation in the way you did?

The experiments are technically sound and the results are presented well. However, it is surprising that the authors did not compare to the RNN/RNN model that they mentioned in the second paragraph of related work. That would seem to be the natural baseline; using the non-preprocessed dataset is a rather trivial baseline, as the performance of recurrent models degrades quite seriously across such long sequences.

**Recommendation.** 4: Ok but not good enough - rejection

**Reasoning.** The Log Signature method has already been shown to be a viable preprocessing technique for time series data. The main contribution of this paper is to show that it also works with CDEs. Since the log signature method is model-independent, it does not seem surprising that this is the case. The experimental results are technically sound but would be improved by adding a stronger baseline. It would make more sense to compare to the RNN/RNN model described in the Related Work section, or to another method of data preprocessing that reduces dimensionality along the time axis.

**To improve the paper.** When using a mathematical concept that will be new to an ML audience, give a one-sentence intuition for what it does and how it is relevant. Try and give a simple, intuitive example of how one might apply the log signature to a short, example time series. Make sure the reader is able to quickly grasp how a log signature of, say, order 2, works in practice. You might consider transitioning to a deeper theoretical treatment after doing so, or possibly refer the reader to a relevant tutorial such as (Chevyrev and Kormilitzina, 2016).

One way to improve the experimental results would be to compare to a stronger baseline such as RNN/RNN. If you don't want to compare to RNN/RNN, you could even just compare to a simple tensor reshape operation that folds the time dimension into the space dimension (eg., given a dataset with axes ["num_examples","time","space] = [a,b * c,d], reshape to [a,b,c * d]). There are probably more clever and effective approaches than that. But the point is that, instead of comparing your dataset preprocessing method to "no dataset preprocessing at all", try comparing it to "another dataset preprocessing method" that does something similar. In this case, that method would involve transferring some of the dataset dimensionality from time to space).

---

> ### Author Response · Authors · 2020-11-12
> **Response**
>
> Thank you for your review.
>
> **Novelty**
> The use of log-signatures is not model-independent. Log-signatures specifically extract information describing how the path drives a controlled differential equation. This has certainly been a motivation for its use in previous work (most notably Liao et al. 2019, who consider RNNs), but it is only with the advent of neural CDE models that this can be taken to its logical conclusion, and actually synergised with a true differential equation model.
>
> In doing so one actually obtains an implementation of the log-ODE method -- rather than just an inspired choice of pre-processing. *This preserves the existing differential equation structure.*
>
> Moreover, previous application of similar ideas have been introduced as general-purpose techniques. Unfortunately, this does not always pan out in practice: performance improvements are often moderate at best, if at all. What previous work has failed to identify, and which we emphasise here, is the importance of application to long time series in particular. Such problems necessarily involve sacrifices which the log-ODE method is well-placed to capitalise upon.
>
> Thus the difference to previous work is we believe significant; we appreciate that this is not completely clear and have added additional commentary to rectify this.
>
>  **Use of theory**
> We have aimed to simplify the theory even further; in particular removing the comparisons to non-ML literature. In detail:
>
> - The term "Riemann--Stieltjes integrals" has been removed. The meaning of the integral is left unambiguous as the path is differentiable;
> - The term "n-fold iterated integral" has been removed as being just terminology;
> - The term "tree-like equivalance" has been replaced with a similar statement on translation invariance;
> - Holder continuity has been replaced with Lipschitz continuity (which is weaker but better-known);
> - The definition of $\beta(v, n)$ (which uses the Mobius function) has been moved to the appendix;
> - The discussion on Lie brackets has been removed, as a more minor point;
> - The comparison to the Magnus expansion has been removed.
>
> We completely recognise the difficulty of these concepts if the theory is unfamiliar. It is precisely because of this that the start of Section 2 ("Theory") begins with *"We begin with motivating theory, though we note that this section is not essential for using the method. Readers more interested in practical applications should feel free to skip to section 3",* and the rest of the paper is careful to be independent of this section. We think this represents a best possible presentation when there is a certain minimum theoretical sophistication that is required, and have now added an additional sentence to help emphasise this.
>
> Beyond even this, we have now used the additional page to introduce further intuition behind these ideas, in particular with respect to the geometry of the input, and the relationship to CDEs. See Section 2.1.
>
>  **Comparison to an RNN Baseline**
> On a serious practical note, RNN-based models do not fit in the memory of the GPU resources we have available. This is one of the main advantages of using differential equation models in the first place, for which adjoint backpropagation is available. (As per the first paragraph of Section 4.)
>
> Indeed one reason we did not compare to an RNN baseline is because they are not ODE (CDE) models. Such methods offer continuous time modelling, adjoint backpropagation, and the ability to easily handle irregular and partially observed time series -- for which this paper represents a demonstration that another kind of difficult data, namely long time series, can also be added to that list. We anticipate that this model will have most utility when one or more of these properties is desired.
>
> We considered reshaping the data as the reviewer suggested, however:
> - Doing so results in incomparable models: the models on reshaped data have dramatically more parameters, due to the cubic scaling of Neural CDE parameter counts. (See section 6.3 of the original Neural CDE paper.) Avoiding this kind of explosion is precisely the purpose of log-signatures as summary statistics.
> - Moreover this typically only makes things worse. The interpolated paths still have roughly the same complexity as before halving the number of data samples -- but now there are two of them. Unlike RNNs, CDEs decouple the number of integration steps from the length of the time series. (A property generally in their favour when considering smoothly-varying data.)
>
> We have added additional commentary to the experimental section to justify our choice.
>
> **Summary**
> We hope we have addressed every concern that the reviewer has raised. We would be very happy to have further discussion if there are any other obstacles to raising the review score.

---

> > ### Comment · AnonReviewer4 · 2020-11-23
> > **Response to authors**
> >
> > Thank you for your thorough and thoughtful response. First, I want to compliment you on the new figure (Figure 2) and on the improved theory section. Your response to my original review was very thoughtful, and I appreciate the clarity with which you wrote it. In what follows, I will continue to be critical of some methods and opinions of the authors, but I want to begin by expressing my respect.
> >
> > Here are areas where I continue to have fundamental disagreement with the authors:
> >
> > * _Model independence._ The authors claim above, "The use of log-signatures is not model-independent." **I emphatically disagree**. If you give me a time series, I can compute the log signature of it and send you the result without ever knowing about neural networks, Neural ODEs, or even such a thing as "Machine Learning." These are orthogonal concepts. It is true that the log signature transform is _compatible_ with a Neural CDE model, but that does not make the log-signature model-dependent. It is a preprocessing operation. The paper does not make this distinction and I believe this to be unintentionally misleading to future readers. Consider the title, "Neural CDES...via the Log-ODE method" -- it implies that we obtain Neural CDEs via this method, which is untrue. A more accurate title would be "Transforming time series with the Log-ODE method improves Neural CDE performance."
> >
> > * _Use of theory._ **I emphatically disagree with this philosophy**: _"We begin with motivating theory, though we note that this section is not essential for using the method. Readers more interested in practical applications should feel free to skip to section 3", and the rest of the paper is careful to be independent of this section."_ I disagree with it because, if the remainder of the paper is independent of these concepts, then _why devote an expository section to them_? When writing any scientific paper, if there is a section which has no bearing on the rest of the paper, then it should be either 1) placed in the appendix or 2) removed entirely. In this case, the theory section should be read by all readers so that they can make sense of the experimental results. It should not be construed to be optional. The good news is that this section is more readable than it was previously.
> >
> > * _Comparison to RNN baseline._ I appreciate the addition of the paragraph beginning in "In principle we could compare against RNN variants..." and I understand how this would make it hard to compare to an RNN variant. Thanks for clarifying! Regarding the reshaping proposition and the authors' response, "the models on reshaped data have dramatically more parameters, due to the cubic scaling of Neural CDE parameter counts.": does this result in a model that is also too large to fit on a GPU? It should not. And if it does, you can decrease the number of hidden units until it does. And it's ok to compare models with a different number of parameters -- most ML papers actually do this to some degree -- so long as you talk about it in the Discussion section and try to make the comparison as fair as possible. I continue to suspect that using "no preprocessing" as a baseline is an unfair comparison.
> >
> > I appreciate the hard work of the authors in making the paper more approachable. Even so, I cannot increase my score because I continue to fundamentally disagree with the authors on the three key points I have discussed.

---

> > > ### Author Response · Authors · 2020-11-23
> > > **Response**
> > >
> > > Thank you for your thoughtful response.
> > >
> > > **Model independence**
> > > This sounds like a small matter of phrasing. Instead of "The use of log-signatures is not model-independent", would the reviewer accept "The *benefit* of log-signatures is not model-independent"?
> > >
> > > We have now adjusted the paper to try and make this distinction clear. (Certainly we do not wish to be misleading.)
> > >
> > > When used as pre-processing for e.g. an RNN model, then the endpoints of the log-signature are fixed. In contrast, we offer a model for which (in the notation of the paper) one can calculate $Z_b$ given $Z_a$ and $X|_{[a, b]}$ *for any choice of $a, b$.* This is a property of being a true ODE model, and not simple pre-processing.
> > >
> > > It is important to note that the alternative suggested title suggested is not accurate. General use of log-signatures is *not* the "log-ODE method". This terms refers specifically to their application, in the manner discussed in the paper, in conjunction with a differential equation.
> > >
> > > **Use of theory**
> > > It is very common for sections of papers to be optional. How often does one skip the details of experiments when reading a paper? It is not that the section "has no bearing on the rest of the paper", it is that it *enhances* the rest of the paper without being a dependency.
> > >
> > > In any case, this is a difference in presentational style. We hope that we can agree to disagree, and respectfully ask that the reviewer not seek to prevent publication on these grounds.
> > >
> > > **Comparison to reshaping**
> > > It is not that it does not fit on the GPU -- but the number of parameters is really very different. Using the EigenWorms dataset as an example, reshaping the channels according to the smallest step size considered (a factor of 8) increases the parameter count from 34629 to 186725; a multiplier of 539%. Even if subsampling is additionally combined with reshaping, so as to only double the number of channels, then the parameter count is increased to 56357; a multiplier of 163%.
> > >
> > > When writing the paper, we (the authors) had a discussion on possible baselines, and concluded that there really were very few sensible options available.

---

### Decision · Program_Chairs · 2021-01-07
**Final Decision**

**Decision:**

Reject

**Comment:**

This paper presents a method for improving the learning of neural controlled differential equation (CDE) models. Neural CDE models provide a number of advantages over neural ODE models in terms of their ability to incorporate continuous-time observations. The primary strength of this paper is that it proposes a mathematically rigorous approach to enable neural CDE models to be learned more efficiently from long time series by converting the CDE to an ODE via the log-ODE method. The results are promising in that the method is able to simultaneously improve accuracy, reduce running time and reduce memory required during learning.

The paper has two main weaknesses. First, the authors claim that due to the problems they are solving (time series with up to 17,000 steps), there are no viable baselines outside of the family of methods that they are proposing. As was noted in the reviews, it would be advisable to consider even very basic baselines for these experiments in addition to current benchmark results. For example, the EigenWorms data set was used in the time series classification benchmark described in Bagnall et al. and there are benchmark results available that appear to outperform those shown in Table 2 (see mean test accuracy results reported here: http://www.timeseriesclassification.com/results/AllAccuracies.zip). The authors are also encouraged to consider even coarse RNN approximations such as partitioning the time series into tractable blocks for learning. It is not clear that the data sets actually have long-range dependencies despite being long.

The second weakness is that the representation that underlies the log-ODE method (the log-signature transform) has been used in previous work in conjunction with discrete-time RNNs. It can be viewed as a preprocessing method in a sense, as was noted by a reviewer. However, it is much more fundamentally integrated with methods for solving CDE's than its prior application to RNNs indicates.

Overall,  support for the paper did not rise to the bar required for acceptance, but we encourage the authors to revise and re-submit the work to a future venue.